# StableVLA: Towards Robust Vision-Language-Action Models without Extra Data

Yiyang Fu [1]  Chubin Zhang [2 3]  Shukai Gong [1]  Yufan Deng [1]  Kaiwei Sun [4]  Qiyang Min [5]  Qibin Hou [6]
Yansong Tang [2]  Jianan Wang [3]  Daquan Zhou [1]

## Abstract

It is infeasible to encompass all possible disturbances within the training dataset. This raises a critical question regarding the robustness of Vision-Language-Action (VLA) models when encountering unseen real-world visual disturbances, particularly under imperfect visual conditions. In this work, we conduct a systematic study based on recent state-of-the-art VLA models and reveal a significant performance drop when visual disturbances absent from the training data are introduced. To mitigate this issue, we propose a lightweight adapter module grounded in information theory, termed the Information Bottleneck Adapter (IB-Adapter), which selectively filters potential noise from visual inputs. Without requiring any extra data or augmentation strategies, IB-Adapter consistently improves over the baseline by an average of 30%, while adding fewer than 10M parameters, demonstrating notable efficiency and effectiveness. Furthermore, even with a $14\times$ smaller backbone (0.5B parameters) and no pre-training on the Open X-Embodiment dataset, our model StableVLA achieves robustness competitive with 7B-scale state-of-the-art VLAs. With negligible parameter overhead ($<$10M), our approach maintains accuracy on long-horizon tasks and surpasses OpenPi under both synthetic and physical visual corruptions. The code is publicly available at https://github.com/DAGroup-PKU/HumanNet/tree/main/src/model/StableVLA.

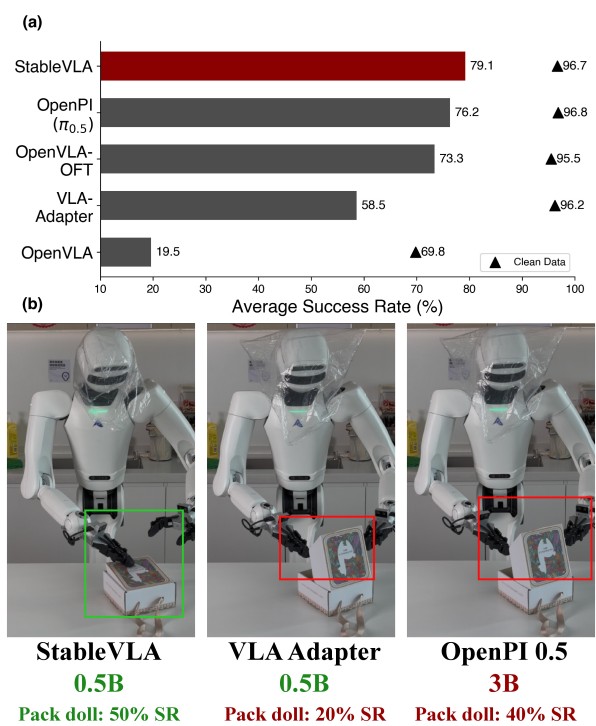

*Figure 1.* **(a) Robustness comparison of StableVLA and other baselines.** Triangle marks and bar charts denote performance on clean data and corrupted data (averaged across severities), respectively. StableVLA achieves state-of-the-art zero-shot robustness. **(b) Real-world robot deployment on the Pack Doll task under visual corruptions.** StableVLA (0.5B) achieves 50% success rate, outperforming both VLA-Adapter (0.5B, 20%) and OpenPI 0.5 (3B, 40%) despite having fewer parameters.

[1]Peking University Shenzhen Graduate School, Beijing, China [2]Tsinghua University, Beijing, China [3]Astribot, Shenzhen, China [4]Nanjing University, Nanjing, China [5]Independent Researcher [6]Nankai University, Tianjin, China. Correspondence to: Daquan Zhou <zhoudaquan21@gmail.com>.

*Proceedings of the 43rd International Conference on Machine Learning*, Seoul, South Korea. PMLR 306, 2026. Copyright 2026 by the author(s).

## 1. Introduction

The integration of Vision–Language Models (VLMs) (Comanici et al., 2025; Bai et al., 2025; Zhu et al., 2025; Xie et al., 2024; Li et al., 2024) into robotic control has fundamentally reshaped the landscape of embodied intelligence. Recent pioneering works (Kim et al., 2024; Zitkovich et al., 2023; Team et al., 2024; Bjorck et al., 2025; Black et al., 2024) demonstrate that effective alignment among visual perception, large language model (LLM) reasoning, and

action execution enables robots to operate across diverse and unstructured scenarios. Building upon this progress, approaches such as VLA-Adapter (Wang et al., 2025) propose efficient mechanisms that bridge vision–language representations to the action space through lightweight policy modules, significantly reducing adaptation overhead. Despite these advances, existing evaluation and benchmarking protocols primarily rely on carefully designed test environments with controlled and idealized visual conditions. In contrast, real-world robotic deployment inevitably involves visual degradations such as sensor noise, motion blur, or weather-induced disturbances, which are largely absent from curated training datasets. This discrepancy introduces a notable gap between model performance observed in benchmark environments and that in real-world settings (Liu et al., 2023a; Mees et al., 2022; Mu et al., 2024). Motivated by this gap, we ask the following question: *How do state-of-the-art VLA models perform when exposed to real-world visual disturbances?* To investigate this, we first evaluate the top-performing VLA-Adapter (Wang et al., 2025) in simulation by injecting synthetic natural visual corruptions. Surprisingly, a model that originally achieved a high success rate of 96% experiences nearly a 50% performance drop under disturbed inputs, as illustrated in Figure 3, and can degrade to 0% success under certain corruption patterns such as severe visual blur. We further demonstrate that this vulnerability is not unique to VLA-Adapter, but also manifests in other leading VLA models, including OpenVLA (Kim et al., 2024), OpenVLA-OFT (Kim et al., 2025), and OpenPi–0.5 (Intelligence et al., 2025). Consistent performance degradation is also observed in real-world experiments conducted with physical robotic systems, as shown in Figure 1 and Table 2.

Prevailing strategies for enhancing robustness primarily rely on using extra data with pre-defined distrubations or data augmentation (Hendrycks et al., 2021) over clean datasets. However, this data-centric approach faces two fundamental limitations. First, simulating the infinite combinatorial space of real-world corruptions is computationally prohibitive. Second, training with augmented data often induces the memorization of specific noise patterns rather than the learning of robust invariant features, which limits generalization ability to unseen corruptions. This raises a pivotal question: *Can we achieve intrinsic robustness through architectural design, without relying on brute-force data scaling?*

We conduct a series of empirical experiments and find evidence suggesting that a significant source of feature vulnerability lies in the projector that bridges the vision encoder and the LLM backbone. As illustrated in Figure 2, substantial feature degradation under noisy visual inputs appears attributable to this projection module. Motivated by the intrinsic feature selection property of the information bottleneck principle (Tishby et al., 2000), we propose a novel block structure for connecting the vision branch and the

LLM backbone, termed IB-Adapter.

By simply replacing the original adapter module in VLA-Adapter and re-training with the same settings, we achieve an average performance improvement of 35.2% across a range of synthetic visual corruptions. In real-robot experiments, our approach yields a 31.7 percentage point improvement in the pick-and-place task. Owing to its strong robustness against visual disturbances, we refer to the resulting model as **StableVLA**. Remarkably, despite this simple architectural replacement and without introducing any additional training data, StableVLA demonstrates robustness that surpasses heavily parameterized baselines, including OpenVLA with $14\times$ more model parameters and $\pi$-0.5 trained with significantly larger amounts of data.

Our contributions are summarized as follows:

- We conduct empirical studies and observe that current state-of-the-art VLA models, despite achieving strong performance on clean benchmark settings, are highly vulnerable to visual disturbances in both synthetic and real-robot scenarios. Furthermore, our analysis provides evidence that this vulnerability is closely associated with the projection module that bridges the vision encoder and the LLM backbone.

- We propose a data-free solution by introducing a novel adapter architecture grounded in information bottleneck theory, termed IB-Adapter. Under zero-shot settings, simply replacing the original adapter with IB-Adapter yields a 35.2% performance improvement over the baseline in the simulator and 20.4 percentage points on real-robot experiments, while keeping all other experimental settings unchanged.

- We conduct extensive experiments across multiple benchmarks, including LIBERO (Liu et al., 2023a), CALVIN (Mees et al., 2022), and real-robot evaluations, on several strong VLA models, such as VLA-Adapter (Wang et al., 2025), OpenVLA (Kim et al., 2024), OpenVLA-OFT (Kim et al., 2025), and $\pi$ (Intelligence et al., 2025). Our results demonstrate that the proposed model consistently outperforms all selected strong baselines while maintaining a significantly smaller model size.

## 2. Related Work and Preliminaries

### 2.1. Robustness in Vision Language Models

Leveraging pre-trained Vision-Language Models (VLMs) (Liu et al., 2023b; Comanici et al., 2025; Liu et al., 2024; Bai et al., 2025; Zhu et al., 2025; Xie et al., 2024; Li et al., 2024) for robotic control has become a dominant paradigm in embodied intelligence (Brohan et al., 2023; Zitkovich et al., 2023; Kim et al., 2024; Team et al., 2024). Training such models from scratch typically depends on massive datasets, including Open

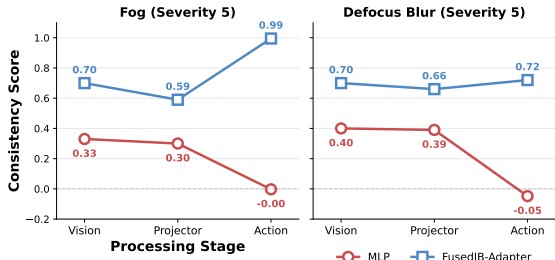

*Figure 2.* Feature consistency across VLA processing stages under visual corruptions (severity 5). The MLP projector shows progressive degradation from Vision to Action, with action predictions collapsing to near-random outputs. Fused IB-Adapter maintains stable representations throughout the pipeline.

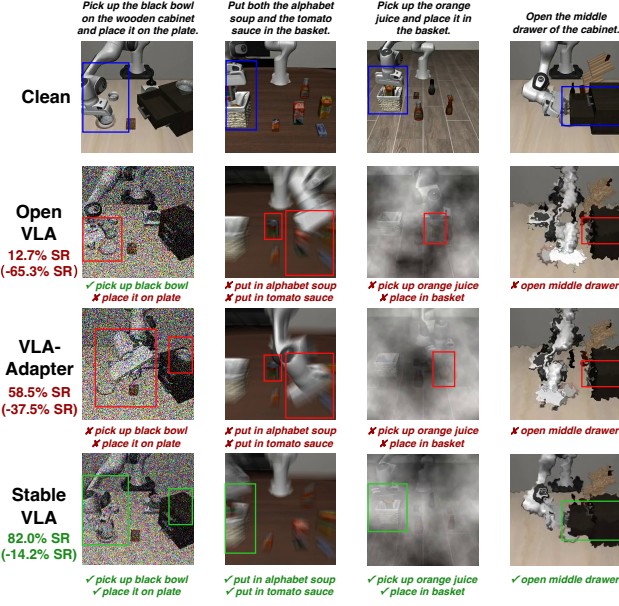

*Figure 3.* **Cases of catastrophic failure.** We evaluate baseline models and StableVLA over noise, blur, weather and digital corruptions at level 5 corruption. While baselines such as OpenVLA and VLA-Adapter exhibit significant performance degradation on corrupted inputs, StableVLA maintains robust performance.

X-Embodiment (O'Neill et al., 2024) and AgiBot (contributors, 2024), and requires substantial computational resources. To alleviate this cost, VLA-Adapter (Wang et al., 2025) introduces a resource-efficient architecture that bypasses large-scale pre-training and directly transfers the general perceptual capabilities of VLMs to robotic domains. However, despite improved training efficiency, a critical challenge remains in architectural robustness. In standard VLA models, the vision encoder (Zhai et al., 2023; Oquab et al., 2023) is commonly frozen to preserve semantic priors (Kim et al., 2024; 2025; Wang et al., 2025), causing input-level noise or corruption to propagate through the visual backbone. Existing approaches rely on simple MLP-based projectors to align visual features with the policy action space, yet such projectors lack intrinsic mechanisms to suppress task-irrelevant disturbances.

Robustness in vision and robotics is traditionally addressed through data-centric strategies, including large-scale data augmentation (Hendrycks & Dietterich, 2019; Hendrycks et al., 2021), and domain randomization in simulation (Tobin et al., 2017). However, these methods are computationally expensive and often fail to generalize to unseen perturbations. To overcome these limitations, we propose **StableVLA**, which targets intrinsic robustness through architectural design by reconstructing the modality alignment interface based on the Information Bottleneck principle, enabling VLA models to effectively filter visual perturbations without relying on exhaustive noise-pattern simulation.

### 2.2. Attention Mechanism from the Perspective of Information Bottleneck

Vision Transformers (ViTs) are more robust to visual corruptions than CNNs (Bai et al., 2021; Paul & Chen, 2022), a property attributed to self-attention, which promotes *visual grouping* by aggregating tokens into semantic clusters (Zhou et al., 2022). This behavior is theoretically grounded in the Information Bottleneck (IB) principle (Tishby et al., 2000), under which self-attention is shown to be equivalent to iterative IB optimization under Gaussian assumptions (Zhou et al., 2022). Beyond spatial attention, channel-wise grouping has been explored through Cross-Covariance Attention in XCiT (Ali et al., 2021) and further interpreted as subspace clustering in FAN (Zhou et al., 2022), where IB-driven channel selection suppresses noise. Building on these insights, **StableVLA** incorporates a multi-head covariance mechanism into VLA modality alignment to filter noisy channels and enable robust semantic propagation.

## 3. Method

In this section, we present **StableVLA**, a framework designed to enhance the intrinsic robustness of VLA models. In Section 3.1, we first formulate the modality alignment problem through the lens of the Information Bottleneck (IB) principle. In Section 3.2, we introduce the idea of **Information Bottleneck Adapter (IB-Adapter)**, which utilizes a channel-wise attention mechanism to suppress visual nuisances while preserving task-relevant semantics. In Section 3.3, we further propose our core contribution, **Fused IB-Adapter**, a hybrid architecture that fuses IB-Adapter with MLP to retain both robust semantics and fine-grained spatial information critical for precise manipulation.

### 3.1. Modality Alignment From an Information Bottleneck Perspective

A standard VLA model typically consists of three main parts: a visual encoder $\mathcal{E}$, a learnable projector $\phi$ for modality alignment, and an LLM-based policy model $\pi$. Given a

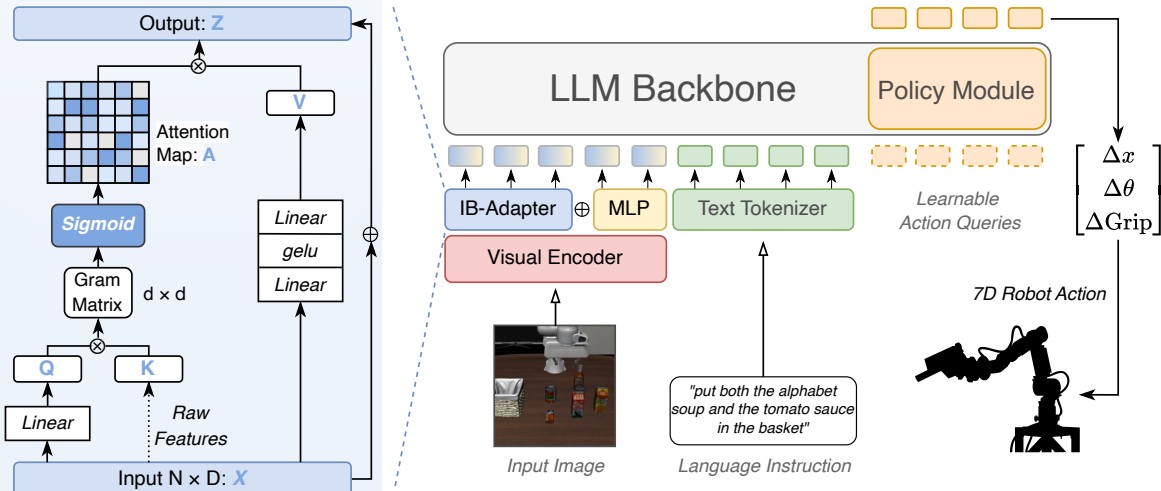

*Figure 4.* **Architecture of the IB-Adapter and StableVLA.** We replace the standard linear bottleneck in StableVLA with Fused IB-Adapter. IB-Adapter processes visual tokens **X** via three parallel pathways to enforce subspace filtering: (1) Covariance Attention: We utilize raw features as keys (dotted line) to preserve original geometric structure. We compute the Gram Matrix ($D \times D$) to capture global channel correlations. (2) Sigmoid Gating: A learnable Sigmoid function generates the Attention Map **A**, acting as an independent gate to suppress noise. (3) Feature transformation: Features undergo non-linear transformation before reconstruction.

visual observation $\mathbf{I}$ and a text instruction $\mathbf{T}$, the encoder extracts visual tokens $\mathbf{X}_v = \mathcal{E}(\mathbf{I}) \in \mathbb{R}^{N \times D_v}$. The projector $\phi$ maps these tokens into the LLM's embedding space: $\mathbf{Z} = \phi(\mathbf{X}_v) \in \mathbb{R}^{N \times D}$. Finally, the LLM predicts actions $\mathbf{a} = \pi(\text{Concat}(\mathbf{Z}, \mathbf{X}_T))$ autoregressively, where $\mathbf{X}_T \in \mathbb{R}^{L \times D}$ represents the text embeddings of $\mathbf{T}$.

In open-world environments, the visual input $\mathbf{I}$ is a composite of task-relevant semantics and task-irrelevant perturbations (e.g., sensor noise). Existing VLA projectors are predominantly implemented as MLP layers. From an Information Bottleneck (IB) perspective, these simple projectors act as *all-pass filters*, which tend to maximize the mutual information $I(\mathbf{X}_v; \mathbf{Z})$ indiscriminately. To enforce intrinsic robustness, we frame modality alignment as an IB problem:

$$\min_{\phi(\mathbf{Z}|\mathbf{X}_v)} \mathcal{L}_{IB} = I(\mathbf{X}_v; \mathbf{Z}) - \beta I(\mathbf{Z}; \mathbf{S}), \quad (1)$$

where $\mathbf{Z}$ is a compressed representation that filters nuisances while retaining the target clean code $\mathbf{S}$ (i.e., the ground-truth task-relevant semantics required to predict actions $\mathbf{a}$). The coefficient $\beta$ controls the trade-off between compression and information preservation.

Crucially, while modern ViT-based encoders effectively leverage the IB-driven grouping mechanism in the spatial dimension (Zhou et al., 2022), we argue that for VLA projectors, performing such grouping across the channel dimension ($D$ features) is more critical for robust alignment. Within the visual encoder's output, semantics and noise are often heterogeneously distributed across channels (Zhou et al., 2022). This motivates our **IB-Adapter**, which treats each channel as an information unit for IB optimization. By

modeling the inter-channel dependencies, IB-Adapter identifies robust semantic subspaces and suppresses uncorrelated noise. Formally,

**Proposition 3.1.** *Let the visual encoder's output* $\mathbf{X}_v = [\mathbf{c}_1, \ldots, \mathbf{c}_D] \in \mathbb{R}^{N \times D}$ *be viewed as a set of $D$ channel-wise observations. Under Gaussian and latent structural assumptions, the iterative update step for the optimal representation* $\mathbf{Z}$ *that minimizes the IB objective in Equation* (1) *corresponds to a channel-wise attention operation:*

$$\mathbf{Z} = \mathbf{V} \cdot \sigma \left( \beta \mathbf{Q}^\top \mathbf{K} \right), \quad (2)$$

*where* $\mathbf{Q}, \mathbf{K}, \mathbf{V} \in \mathbb{R}^{N \times D}$ *are linear projections of* $\mathbf{X}_v$. *The operator* $\sigma(\cdot)$ *is a normalizer determined by the latent distribution assumption:* $\sigma(\cdot)$ *takes the form of **Softmax** under a categorical latent structure, or **Sigmoid** under an independent Bernoulli latent structure.*

The detailed derivation is provided in Appendix A. By extending IB-driven grouping to the channel dimension, this approach enables adaptive filtering to dynamically suppress noisy channels while accentuating stable features, thereby establishing representation robustness before features are propagated into the downstream policy model.

### 3.2. Information Bottleneck Adapter (IB-Adapter)

To enforce the IB principle within the modality alignment stage, we propose the Information Bottleneck Adapter (IB-Adapter). Unlike MLPs that process channels independently, IB-Adapter models the inter-channel covariance to identify and amplify robust semantic signals.

Let $\mathbf{X}' \in \mathbb{R}^{N \times D}$ be the input features (e.g., intermediate projector features). The mechanism consists of three critical components: subspace covariance modeling, sigmoid-based gating, and non-linear feature transformation.

**Subspace covariance modeling.** We adopt a multi-head design to capture correlations across $H$ diverse semantic subspaces. The input $\mathbf{X}' \in \mathbb{R}^{N \times D}$ is partitioned into $H$ heads $[\mathbf{X}'_1, \ldots, \mathbf{X}'_H]$, where each head $\mathbf{X}'_h \in \mathbb{R}^{N \times d}$ has a channel dimension $d = D/H$. For each head $h$, we derive the query $\mathbf{Q}_h = \mathbf{X}'_h \mathbf{W}_q \in \mathbb{R}^{N \times d}$ through a learnable projection $\mathbf{W}_q \in \mathbb{R}^{d \times d}$, while the key $\mathbf{K}_h = \mathbf{X}'_h \in \mathbb{R}^{N \times d}$ is defined via an identity mapping of the input features. This identity-key design ensures that the subsequent covariance computation is grounded in the intrinsic geometric manifold of the visual tokens, thereby preserving high-frequency spatial cues that might otherwise be attenuated by redundant projections. To model inter-channel dependencies, we compute a Gram matrix $\mathbf{G}_h$ by aggregating correlations along the sequence dimension $N$:

$$\mathbf{G}_h = \mathbf{Q}_h^\top \mathbf{K}_h \in \mathbb{R}^{d \times d}, \tag{3}$$

where each element $\mathbf{G}_h[i, j]$ represents the covariance between channel $i$ and channel $j$ across all spatial tokens.

**Sigmoid-based subspace gating.** To separate semantic clusters from independent noise, we apply a learnable sigmoid gating function to the Gram matrix:

$$\mathbf{A}_h = \sigma\left(\mathbf{G}_h \cdot \boldsymbol{\tau}_h\right) \in [0,1]^{d \times d}, \tag{4}$$

where $\boldsymbol{\tau}_h$ is a learnable temperature parameter. The use of sigmoid gating function is theoretically motivated by the *independent Bernoulli latent structure assumption* of the channels. A channel representing uncorrelated sensor noise should exhibit low covariance with semantic-bearing channels, resulting in a gate value near zero. Unlike Softmax operation, which enforces competition between channels by enforcing a categorical distribution over channels, sigmoid gating allows for independent channel selection by suppressing such noisy channels independently without affecting the energy of robust semantic channels.

**Non-linear feature transformation.** To enhance feature expressivity, the input $\mathbf{X}_h \in \mathbb{R}^{N \times d}$ is transformed via a two-layer MLP with GELU activation to generate the value tokens $\mathbf{V}_h \in \mathbb{R}^{N \times d}$:

$$\mathbf{V}_h = \text{Norm}(\text{GELU}(\mathbf{X}_h \mathbf{W}_{v1}) \mathbf{W}_{v2}), \tag{5}$$

where $\mathbf{W}_{v1}, \mathbf{W}_{v2} \in \mathbb{R}^{d \times d}$ are learnable weights. The head output $\mathbf{Z}_h \in \mathbb{R}^{N \times d}$ is then reconstructed by modulating these features with the spectral gate $\mathbf{A}_h \in \mathbb{R}^{d \times d}$:

$$\mathbf{Z}_h = \mathbf{V}_h \mathbf{A}_h, \tag{6}$$

and thus $\mathbf{Z} = [\mathbf{Z}_1, \cdots, \mathbf{Z}_H]$. IB-Adapter couples non-linear synthesis with channel-wise noise suppression. This design satisfies the IB compression objective (Equation (1)) by filtering out visual nuisances before the representations are propagated to the LLM backbone.

### 3.3. Hybrid Architecture for Balancing Robust Semantics and High-frequency Details

While IB-Adapter effectively suppresses visual disturbance and promotes semantic robustness, it can attenuate high-frequency details essential for precise manipulation. This challenge is particularly evident in long-horizon tasks, where trajectory precision must be maintained over extended sequences. To resolve this trade-off, we propose **Fused IB-Adapter**, a dual-pathway architecture designed to decouple robust semantic understanding from precise spatial execution:

$$\mathbf{Z} = \text{MLP}(\mathbf{X}) + \tanh(\lambda) \cdot \text{IB-Adapter}(\mathbf{X}), \tag{7}$$

where $\lambda$ controls the injection of robust signals. This design maintains two parallel pathways: $i$) a high-fidelity path using a standard MLP to preserve raw high-frequency details essential for fine-motor control, and $ii$) a denoising path using the IB-Adapter module to extract robust, covariance-filtered semantic features.

To tailor this balance to specific task dynamics, we calibrate the **Stochastic Pathway Dropout (SPD)** rate $p_{\text{drop}}$ during fine-tuning. For tasks demanding extreme spatial fidelity for pick-and-place operations (e.g., LIBERO-Long), retaining the MLP pathway ($p_{\text{drop}} \approx 0$) is crucial. In these scenarios, the IB-Adapter acts as a robustness residual, stabilizing the representation without sacrificing the high-frequency cues required for precise execution. For tasks requiring consistent object identification or long-horizon semantic planning(e.g., CALVIN, LIBERO-Object), a moderate dropout ($p_{\text{drop}} \approx 0.3$) forces the policy to internalize the robust features from the IB pathway, preventing semantic drift under visual corruptions. This task-specific configuration allows StableVLA to flexibly navigate the robustness landscape across diverse robotic domains.

## 4. Experiments

### 4.1. Results on Benchmark

#### 4.1.1. SETUP

**Benchmarks.** We select the widely adopted LIBERO (Liu et al., 2023a) to evaluate the performance of StableVLA on various types of tasks, and select CALVIN (Mees et al., 2022) benchmark to evaluate the zero-shot generalization of StableVLA. For LIBERO, we utilize all four task categories: LIBERO-Spatial, LIBERO-Object, LIBERO-Goal, and LIBERO-Long. Each task suit contains 10 subtasks,

*Table 1.* Full comparison on LIBERO and CALVIN benchmark. Methods are grouped by training paradigm. We report success rate (%). **Bold**: best, underline: second best.

| Training Method | Method | Spatial | | | | Object | | | | Goal | | | | Long | | | | CALVIN | | | |
|---|---|---|---|---|---|---|---|---|---|---|---|---|---|---|---|---|---|---|---|---|---|
| | | C | S3 | S4 | S5 | C | S3 | S4 | S5 | C | S3 | S4 | S5 | C | S3 | S4 | S5 | C | S3 | S4 | S5 |
| OpenX Pretrain | OpenVLA (7B) | 80.0 | 40.9 | 24.6 | 14.7 | 69.6 | 18.2 | 10.4 | 2.7 | 74.0 | 38.7 | 27.0 | 16.3 | 55.5 | 20.5 | 12.4 | 7.0 | – | – | – | – |
| | OpenVLA-OFT (7B) | 92.6 | 89.3 | 84.0 | 72.1 | 98.4 | 82.5 | 69.2 | 52.8 | 96.8 | **94.5** | 84.6 | 70.3 | **94.4** | **77.6** | 61.9 | 40.3 | – | – | – | – |
| OpenX+Web Co-train | OpenPi–0.5 (3B) | **98.4** | 88.3 | 79.0 | 62.4 | **99.4** | **97.1** | **88.4** | **76.4** | 97.2 | 87.2 | 82.5 | 64.2 | 92.0 | 76.1 | **65.6** | 47.7 | – | – | – | – |
| VLM Direct FT | VLA-Adapter (0.5B) | 96.0 | 93.7 | 83.3 | 58.5 | 96.8 | 71.0 | 44.1 | 29.3 | 97.4 | 79.5 | 64.7 | 47.3 | 94.4 | 63.5 | 41.0 | 26.2 | 4.14 | 2.56 | 1.89 | 1.44 |
| | StableVLA (0.5B) | 96.2 | **94.4** | **92.1** | **82.0** | 98.8 | 92.4 | 83.6 | 70.2 | **98.0** | 93.4 | 85.0 | 71.9 | 93.6 | 76.3 | 62.4 | 45.3 | **4.17** | **2.77** | **2.11** | **1.51** |

where each subtask is repeated for 50 episodes for evaluation. We report the averaged success rate (ranging from 0 to 100%) over all 500 episodes for each task suit. For CALVIN, StableVLA is evaluated on environment unseen during training to test its generalization performance. Specifically, StableVLA is required to execute a predefined sequence of 1,000 tasks in order. Each individual task is composed of five subtasks, and the model may only move on to the subsequent subtask once the current one has been completed. We report the average completed tasks (ranging from 0 to 5).

**Corruption Protocol.** To rigorously evaluate intrinsic robustness, we adopt the corruption protocol from ImageNet-C (Hendrycks & Dietterich, 2019). We utilize the comprehensive set of corruptions provided by the imagecorruptions library (Michaelis et al., 2019), spanning four categories: noise, blur, weather, and digital corruptions.[1] These corruptions are defined across 5 severity levels, and we focus our evaluation on the challenging high-severity regime (Levels 3–5) to stress-test architectural stability. For all evaluations, we conduct experiments over distinct intensity settings (clean, plus severity levels 3, 4, and 5) across the deployed corruption types.

**Training Protocol.** StableVLA replaces the MLP projectors in the VLA-Adapter (Wang et al., 2025) framework with our Fused IB-Adapter module and is trained from scratch on LIBERO and CALVIN. Detailed hyperparameters, infrastructure specifications, and baseline configurations are provided in Appendix B. Following standard VLA training recipes, we apply mild geometric (crop) and photometric (color jitter) augmentations during training to prevent overfitting. Crucially, we do not expose the models to any aforementioned corruptions or use any specialized robustness techniques (e.g., data augmentation). Thus, the evaluation on the corruptions remains a strictly zero-shot test of architectural generalization.

---

[1] We evaluate the full spectrum of 19 corruptions on LIBERO-Spatial. For LIBERO-Object/Goal/Long and CALVIN benchmarks, we exclude *Glass Blur* due to its prohibitive computational cost during interaction, resulting in 18 corruption types for these tasks.

**Baselines.** We benchmark against the state-of-the-art VLA-Adapter (Wang et al., 2025), along with VLAs such as OpenVLA (Kim et al., 2024), OpenVLA-OFT (Kim et al., 2025) and OpenPi–0.5 (Intelligence et al., 2025).

4.1.2. EXPERIMENT RESULTS.

**Comprehensive Robustness Profile.** In Table 1, we report per-task success rates for LIBERO and average completed tasks for CALVIN, with both metrics averaged over all corruption types. StableVLA demonstrates superior zero-shot robustness across all benchmarks and corruption severity levels. On LIBERO, StableVLA consistently scores best or second best across all baselines, rivaling OpenVLA-OFT and OpenPi–0.5 pretrained on large scale datasets like OpenX-Embodiment. It achieves notable improvements over VLA-Adapter that shares similar architecture, especially under severe corruptions, with performance improvements of 40.2% to 139.6% across 4 task suites at severity level 5. On CALVIN, StableVLA consistently completes more tasks than VLA-Adapter across all corruption levels. These results demonstrate the superior zero-shot robustness of StableVLA across various manipulative tasks. More detailed results are provided in Appendix C.

As illustrated in Figure 5, StableVLA demonstrates a comprehensive robustness advantage, surpassing the VLA-Adapter baseline across the vast majority of corruption categories in all four task suites. Notably, despite the parameter disparity, StableVLA achieves robustness levels competitive with or surpassing large-scale SOTA models (OpenPi– and OpenVLA-OFT). This supports the hypothesis that the IB-Adapter module effectively extracts robust semantic features across diverse manipulation scenarios, bridging the performance gap with significantly larger foundation models.

**Visualization of Fused IB-Adapter features.** To demystify the mechanism behind the quantitative gains, we conducted semantic clustering visualization to visualize the features of Fused IB-Adapter . Specifically, we conduct K-Means clustering ($K = 2$) to the output features of MLP in VLA-Adapter and Fused IB-Adapter in StableVLA. For illustration purpose, we examine a manipulation scene from

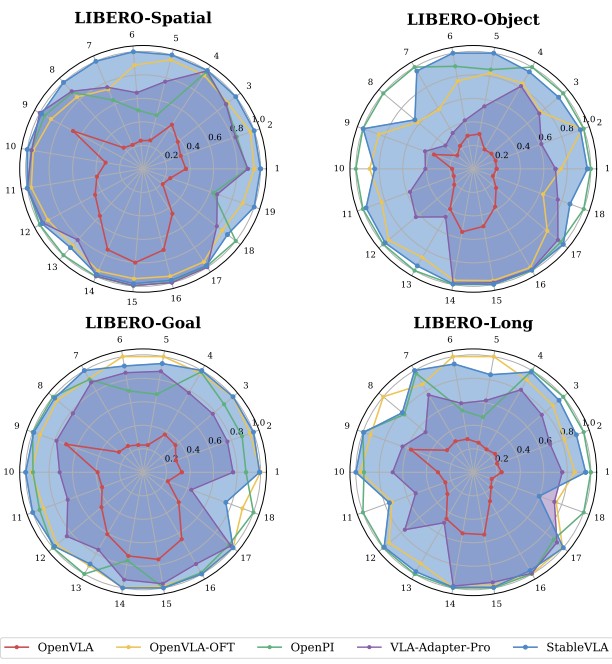

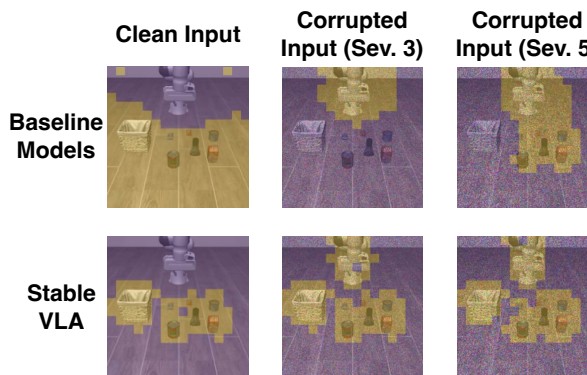

*"Pick up the alphabet soup and place it in the basket"*

*Figure 6.* **Semantic grouping of Fused IB-Adapter output features.** We injected speckle noise at severity levels 3 and 5 into the input image of a LIBERO-Object task ("Pick up the alphabet soup and place it in the basket"). Then, we applied K-means clustering (K=2) to visualize and compare the semantic clusters in the output features across clean, level-3 and level-5 noisy inputs. The comparison between baseline models and Stable VLA demonstrates that StableVLA maintains more consistent and accurate semantic grouping even under high levels of noise corruption.

*Figure 5.* Robustness comparison across visual corruption types on LIBERO. Each axis represents a corruption type. See Appendix D for details.

the LIBERO-Object suite under impulse noise, a corruption type characterized by high-frequency, spatially independent disturbances. Figure 6 demonstrates that even without noise (left column), standard MLPs produce diffused features that conflate task-relevant regions with background. This effect is further amplified under high noise (right columns), which suggests that the standard projector propagates high-frequency disturbances to the downstream policy. In contrast, the output of Fused IB-Adapter produces coherent, object-centric semantic groupings. We attribute this to the covariance-based Sigmoid gating in Fused IB-Adapter. Stochastic corruptions exhibit low correlation with object structures, producing low Gram matrix values that the Sigmoid gates suppress, resulting in a clean focus on grippers and manipulation targets.

## 4.2. Real-world Robot Deployment

### 4.2.1. EXPERIMENT SETUP

**Robot Platform.** We conduct real-world experiments using the Astribot S1, a high-precision dual-arm robot platform (Gao et al., 2025). The robot's chassis and torso are immobilized; control is restricted to the 14-DoF dual arms and gripper actuation. Sensory input consists of RGB streams from a head-mounted camera, which tracks the workspace center, and two wrist-mounted cameras.

**Tasks and Data.** We designed four distinct tasks to evaluate various facets of robotic capability, ranging from basic

manipulation to long-horizon planning. A visualization of the task processes is provided in Figure 8.

1. **Pick and Place:** Evaluates basic manipulation proficiency. We utilize a set of 5 different objects for comprehensive evaluation, with each object tested over 2 trials. We collected 1000 teleoperated demonstrations for fine-tuning.

2. **Throw Basketball:** Evaluates manipulation skills for small objects. We collected 500 teleoperated demonstrations for this task.

3. **Pour Water:** Evaluates manipulation precision. We collected 500 teleoperated demonstrations for this task.

4. **Pack the Doll:** A long-horizon task requiring multi-stage planning: picking up a doll, placing it precisely into a box, and closing the lid. This tests the model's ability to handle precise geometric constraints. We collected 200 teleoperated demonstrations for fine-tuning.

All expert demonstrations were acquired via VR teleoperation, with the head camera actively tracking the workspace center. Each model is evaluated over 10 trials per task under each corruption setting.

**Benchmark Protocol.** For our real-world experiments, we introduce four types of visual corruptions. First, we simulate two digital corruptions—**Gaussian noise** and **defocus blur**—using the `imagecorruptions` library, applying these effects to the input images prior to model inference.

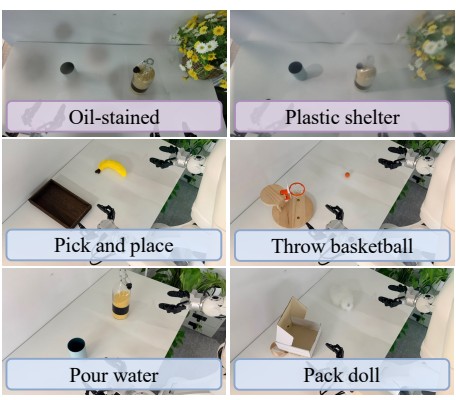

*Figure 7.* **Visualization of real-world setup.** Top: Physical corruptions (Oil and Shelter). Bottom: Initial states of four evaluation tasks.

*Table 2.* **Real-world robustness evaluation.** "Clean" reports the absolute success rate (%). All other columns report the **performance change** (percentage points, $\Delta$) relative to the clean setting. **Noise** and **Blur** represent the average performance drop across different severity levels. *Oil* and *Shelter* denote physical interferences. **Bold** indicates the best robustness (smallest performance drop).

| Task | Method | Clean | Noise ($\Delta$) | Blur ($\Delta$) | *Oil* ($\Delta$) | *Shelter* ($\Delta$) | Avg. ($\Delta$) |
|---|---|---|---|---|---|---|---|
| Pick and place | $\pi_{0.5}$ | 100.0 | -63.3 | -16.7 | -10.0 | -30.0 | -30.1 |
| | VLA-Adapter | 80.0 | -66.7 | -40.0 | -30.0 | -60.0 | -49.2 |
| | **StableVLA** | 80.0 | **-30.0** | **-10.0** | **-10.0** | **-20.0** | **-17.5** |
| Throw basketball | $\pi_{0.5}$ | 80.0 | -60.0 | -33.3 | -20.0 | -30.0 | -35.8 |
| | VLA-Adapter | 60.0 | -53.0 | -40.0 | -20.0 | -40.0 | -38.3 |
| | **StableVLA** | 60.0 | **-36.7** | **-16.7** | **-10.0** | **-10.0** | **-18.4** |
| Pour water | $\pi_{0.5}$ | 70.0 | -60.0 | -20.0 | -20.0 | -20.0 | -30.0 |
| | VLA-Adapter | 40.0 | -40.0 | -30.0 | -10.0 | -20.0 | -25.0 |
| | **StableVLA** | 40.0 | **-23.3** | **-16.7** | **-0.0** | **-10.0** | **-12.5** |
| Pack doll | $\pi_{0.5}$ | 80.0 | -63.3 | -33.3 | -30.0 | -40.0 | -41.7 |
| | VLA-Adapter | 50.0 | -40.0 | -26.7 | -30.0 | -30.0 | -31.7 |
| | **StableVLA** | 60.0 | **-16.7** | **-10.0** | **-20.0** | **-10.0** | **-14.2** |

*Table 3.* Ablation study on adapter architecture. We report success rate (%) for LIBERO (averaged across 4 task suites) and average completed tasks for CALVIN. **Bold**: best.

| Benchmark | Method | Clean | Avg |
|---|---|---|---|
| LIBERO | IB-Adapter | 96.3 | 76.0 |
| | Fused IB-Adapter | **96.6** | **79.1** |
| | Fused IB-Adapter (Softmax) | 89.6 | 62.8 |
| CALVIN | IB-Adapter | 1.64 | 1.44 |
| | Fused IB-Adapter | **4.17** | **2.13** |
| | Fused IB-Adapter (Softmax) | 0.46 | 0.46 |

We evaluate Gaussian noise at severity levels 2–4 and defocus blur at levels 3–5, as preliminary experiments indicated that high-severity Gaussian noise leads to catastrophic performance degradation across all baselines. Second, we introduce physical distortions by directly obstructing the camera lens with **oil** and a **plastic cover**, separately. The resulting corrupted images are shown in Fig. 7. For a fair comparison, we benchmark against $\pi_{0.5}$ (Intelligence et al., 2025), a strong generalist VLA model pretrained on large-scale embodied data, and VLA-Adapter (Wang et al., 2025), which is trained exclusively on our collected dataset.

4.2.2. EXPERIMENT RESULTS

As shown in Table 2, while the generalist baseline $\pi_{0.5}$ achieves high success rates on clean data, it suffers significant performance degradation under visual corruptions. VLA-Adapter exhibits similar fragility. In contrast, StableVLA demonstrates superior robustness, consistently maintaining the smallest performance drop across all tasks. Notably, our method shows exceptional resilience against physical interferences (Oil and Shelter), effectively preserving manipulation capabilities where baselines falter.

### 4.3. Ablation Studies

We validate two core design of IB-Adapter in Table 3.

**Dual-Stream Necessity.** As shown in Table 3, removing the high-fidelity MLP pathway results in performance degradation. On LIBERO, the average success rate of IB-Adapter is 3.1% lower than that of Fused IB-Adapter on corrupted data. On CALVIN, the average number of completed tasks falls from 2.14 to 1.44 when switching from Fused IB-Adapter to IB-Adapter. The dual-stream design proves to be critical: the MLP pathway preserves fine-grained spatial details, while the IB pathway provides semantic robustness, synergizing to achieve optimal performance.

**Sigmoid vs. Softmax.** As shown in Table 3, replacing Sigmoid with Softmax leads to a significant performance drop across all benchmarks. On LIBERO, the Softmax variant of Fused IB-Adapter witnesses a 26.8% drop on corrupted data. On CALVIN, the average completed tasks collapses from 2.13 to 0.46 on corrupted data. This empirically confirms our independent Bernoulli latent structure assumption proposed in Section 3.2. Unlike Softmax which enforces competition, Sigmoid activation enables independent channel suppression, acting as a more effective filter for noise.

## 5. Conclusions

In this work, we addressed the notable vulnerability of VLA modality alignment to visual corruptions. Guided by the Information Bottleneck principle, we proposed the **IB-Adapter**, a general-purpose dual-stream architecture that synergizes high-fidelity spatial processing with covariance-based spectral filtering. Extensive evaluations confirm that this design confers superior zero-shot robustness across diverse benchmarks. Remarkably, our results demonstrate that **architectural innovation can help narrow the scaling gap**: despite using a $14\times$ smaller backbone without external OpenX pre-training, our approach achieves robustness competitive with data-intensive 7B-scale SOTA models. As a parameter-efficient and potentially **model-agnostic** component, the IB-Adapter offers a compelling alternative

to standard projectors, encouraging a shift towards robust architectural inductive biases in the development of next-generation embodied AI.

## Acknowledgements

We gratefully acknowledge Simple Silicon Innovation for their valuable support throughout this project. We also thank the National Natural Science Foundation of China (NSFC) for partially supporting Qibin Hou under Grant No. 62522607.

## Impact Statement

This paper presents work whose goal is to advance the field of Machine Learning. There are many potential societal consequences of our work, none of which we feel must be specifically highlighted here.

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

## A. Theoretical Derivation

In this section, we provide a rigorous derivation of Proposition 3.1. Let $\mathbf{X} = [\mathbf{c}_1, \ldots, \mathbf{c}_D] \in \mathbb{R}^{N \times D}$ be the visual encoder output, where each column $\mathbf{c}_j \in \mathbb{R}^N$ represents a channel-wise feature vector across $N$ spatial tokens. Our goal is to find an optimal compressed representation $\mathbf{Z}$ that minimizes the IB Lagrangian:

$$\mathcal{L} = I(\mathbf{X}; \mathbf{Z}) - \beta I(\mathbf{Z}; \mathbf{S}), \tag{8}$$

where $\mathbf{S}$ is the target clean code. Following the IB framework for unsupervised clustering, we treat each channel $\mathbf{c}_j$ as a data point (indexed by $j$) to be clustered into $D$ semantic groups (indexed by $c$). We assume the data distribution follows $p(\mathbf{s}|j) = \mathcal{N}(\mathbf{s}|\mathbf{c}_j, \epsilon^2 \mathbf{I})$ where $\epsilon$ is a smoothing parameter. The $t$-th iterative step for the soft assignment of channel $j$ to cluster $c$ is given by (Tishby et al., 2000):

$$q^{(t)}(c|j) = \frac{p^{(t-1)}(c)}{Z(\mathbf{c}_j, \beta)} \exp\left[-\beta D_{KL}[p(\mathbf{s}|j) \| p^{(t-1)}(\mathbf{s}|c)]\right], \tag{9}$$

$$p^{(t)}(c) = \frac{n_c^{(t)}}{D},$$

where $Z(\mathbf{c}_j, \beta)$ is the partition function. We approximate the cluster-conditional distribution $p(\mathbf{s}|c)$ with a Gaussian $g(\mathbf{s}|c) = \mathcal{N}(\mathbf{s}|\boldsymbol{\mu}_c, \Sigma_c)$ and assume $\epsilon$ is sufficiently small. The KL divergence between $p(\mathbf{s}|j)$ and $g(\mathbf{s}|c)$ possesses the closed form:

$$D_{KL}[p(\mathbf{s}|j) \| g(\mathbf{s}|c)] = \frac{1}{2}\left[\log\frac{|\Sigma_c|}{|\epsilon^2 \mathbf{I}|} - N + \operatorname{tr}(\Sigma_c^{-1}\epsilon^2 \mathbf{I}) + (\mathbf{c}_j - \boldsymbol{\mu}_c)^\top \Sigma_c^{-1}(\mathbf{c}_j - \boldsymbol{\mu}_c)\right]$$

$$\xrightarrow{\epsilon \to 0} \frac{1}{2}\left[\log|\Sigma_c| + (\mathbf{c}_j - \boldsymbol{\mu}_c)^\top \Sigma_c^{-1}(\mathbf{c}_j - \boldsymbol{\mu}_c)\right] + \text{const}$$

$$\propto (\mathbf{c}_j - \boldsymbol{\mu}_c)^\top \Sigma_c^{-1}(\mathbf{c}_j - \boldsymbol{\mu}_c) + \log|\Sigma_c|. \tag{10}$$

Plugging Equation (10) into Equation (9), we obtain:

$$q^{(t)}(c|j) = \frac{n_c^{(t-1)}/D}{|\Sigma_c|^{\beta/2}} \frac{\exp\left(-\frac{\beta}{2}(\mathbf{c}_j - \boldsymbol{\mu}_c)^\top \Sigma^{-1}(\mathbf{c}_j - \boldsymbol{\mu}_c)\right)}{Z(\mathbf{c}_j, \beta)},$$

where the constant terms are absorbed into the $Z(\mathbf{c}_j, \beta)$. Looking into the quadratic form $(\mathbf{c}_j - \boldsymbol{\mu}_c)^\top \Sigma^{-1}(\mathbf{c}_j - \boldsymbol{\mu}_c)$, for a fixed $j$, the term $\mathbf{c}_j^\top \Sigma^{-1} \mathbf{c}_j$ is constant across all clusters $c$ and can be absorbed into $Z(\mathbf{c}_j, \beta)$. Without loss of generality, we assume a shared covariance $\Sigma_c = \Sigma$ across clusters and normalized cluster centers such that $\boldsymbol{\mu}_c^\top \Sigma^{-1} \boldsymbol{\mu}_c = 1$, we have:

$$q^{(t)}(c|j) = \frac{n_c^{(t-1)}/D}{|\Sigma|^{\beta/2}} \frac{\exp\left(\beta \boldsymbol{\mu}_c^\top \Sigma^{-1} \mathbf{c}_j\right)}{Z(\mathbf{c}_j, \beta)}.$$

The specific form of the partition function $Z(\mathbf{c}_j, \beta)$ determines the normalization and the resulting attention mechanism. We now consider two cases based on different assumptions about the latent structure.

**Case 1: Categorical Latent Structure.** Under the assumption that each channel $j$ must be assigned to *exactly one* cluster, the assignments form a categorical distribution. The partition function enforcing this constraint is $Z(\mathbf{c}_j, \beta) = \sum_c \exp\left(\beta \boldsymbol{\mu}_c^\top \Sigma^{-1} \mathbf{c}_j\right)$, leading to:

$$q^{(t)}(c|j) = \frac{n_c^{(t-1)}/D}{|\Sigma|^{\beta/2}} \frac{\exp\left(\beta \boldsymbol{\mu}_c^\top \Sigma^{-1} \mathbf{c}_j\right)}{\sum_c \exp\left(\beta \boldsymbol{\mu}_c^\top \Sigma^{-1} \mathbf{c}_j\right)}.$$

We define the output representation $\mathbf{z}_c^{(t)}$ as these updated cluster centers, i.e.

$$\mathbf{z}_c^{(t)} := \boldsymbol{\mu}_c^{(t)} = \frac{1}{n_c^{(t)}} \sum_{j=1}^{D} q^{(t)}(c|j)\mathbf{c}_j = \sum_{j=1}^{D} \frac{n_c^{(t-1)}/D}{n_c^{(t)}|\Sigma|^{\beta/2}} \frac{\exp\left(\beta \boldsymbol{\mu}_c^\top \Sigma^{-1} \mathbf{c}_j\right)}{\sum_c \exp\left(\beta \boldsymbol{\mu}_c^\top \Sigma^{-1} \mathbf{c}_j\right)} \tag{11}$$

$$= \sum_{j=1}^{D} \frac{\exp\left(\beta \mathbf{k}_c^\top \mathbf{q}_j\right)}{\sum_c \exp\left(\beta \mathbf{k}_c^\top \mathbf{q}_j\right)} \mathbf{v}_j = \sum_{j=1}^{D} \operatorname{Softmax}_c(\beta \mathbf{k}_c^\top \mathbf{q}_j)\mathbf{v}_j, \tag{12}$$

where we define $\mathbf{q}_j = \Sigma^{-1}\mathbf{c}_j$, $\mathbf{k}_c = \boldsymbol{\mu}_c^{(t-1)}$ and $\mathbf{v}_j = \frac{n_c^{(t-1)}/D}{n_c^{(t)}|\Sigma|^{\beta/2}}\mathbf{c}_j$. In matrix form, this yields,

$$\mathbf{Z} = \mathbf{V} \cdot \mathrm{Softmax}\left(\beta\mathbf{Q}^\top\mathbf{K}\right), \tag{13}$$

where $\mathbf{Q} = \Sigma^{-1}[\mathbf{c}_1,\ldots,\mathbf{c}_D] = \mathbf{W}_Q\mathbf{X}$, $\mathbf{K} = [\boldsymbol{\mu}_1^{(t-1)},\ldots,\boldsymbol{\mu}_D^{(t-1)}] = \mathbf{W}_K\mathbf{X}$, $\mathbf{V} = \frac{n_c^{(t-1)}/D}{n_c^{(t)}|\Sigma|^{\beta/2}}[\mathbf{c}_1,\ldots,\mathbf{c}_D] = \mathbf{W}_V\mathbf{X}$, $\mathbf{Z} = [\boldsymbol{\mu}_1^{(t)},\ldots,\boldsymbol{\mu}_D^{(t)}]$, and $b$ is a learnable bias. Here, $\mathbf{W}_Q, \mathbf{W}_K, \mathbf{W}_V$ are learnable parameters.

**Case 2: Independent Bernoulli Latent Structure.** We now relax the categorical constraint. Instead of requiring each channel to be assigned to exactly one semantic group, we assume that the association of channel $j$ with each cluster $c$ is an *independent* binary decision. Let $a_{jc} \in \{0, 1\}$ be a binary latent variable where $a_{jc} = 1$ denotes that channel $j$ is associated with cluster $c$, following an independent Bernoulli distribution. For a specific pair $(j, c)$, the local partition function over the binary state space $\{0, 1\}$ is:

$$Z(\mathbf{c}_j, \beta) = \exp\left(\beta\boldsymbol{\mu}_c^\top\Sigma^{-1}\mathbf{c}_j\right) + \exp(b), \tag{14}$$

where $b$ is a learnable bias representing the activation threshold (the "off" state energy). Under this formulation, the probability that channel $j$ carries the semantic information of cluster $c$ is:

$$q^{(t)}(a_{jc} = 1|j) = \frac{n_c/D}{|\Sigma|^{\beta/2}} \frac{\exp\left(\beta\boldsymbol{\mu}_c^\top\Sigma^{-1}\mathbf{c}_j\right)}{\exp\left(\beta\boldsymbol{\mu}_c^\top\Sigma^{-1}\mathbf{c}_j\right) + \exp(b)} \tag{15}$$

Similar to the derivation in Case 1, we define the output representation $\mathbf{z}_c^{(t)}$ as these updated cluster centers, i.e.

$$\mathbf{z}_c^{(t)} := \boldsymbol{\mu}_c^{(t)} = \frac{1}{n_c^{(t)}}\sum_{j=1}^D q^{(t)}(a_{jc} = 1|j) \cdot \mathbf{c}_j = \sum_{j=1}^D \frac{n_c^{(t-1)}/D}{n_c^{(t)}|\Sigma|^{\beta/2}} \frac{\exp\left(\beta\boldsymbol{\mu}_c^\top\Sigma^{-1}\mathbf{c}_j\right)}{\exp\left(\beta\boldsymbol{\mu}_c^\top\Sigma^{-1}\mathbf{c}_j\right) + \exp(b)}\mathbf{c}_j \tag{16}$$

$$= \sum_{j=1}^D \frac{\exp\left(\beta\mathbf{k}_c^\top\mathbf{q}_j\right)}{\exp\left(\beta\mathbf{k}_c^\top\mathbf{q}_j\right) + \exp(b)}\mathbf{v}_j = \sum_{j=1}^D \sigma(\beta\mathbf{k}_c^\top\mathbf{q}_j - b)\mathbf{v}_j, \tag{17}$$

where we define $\mathbf{q}_j = \Sigma^{-1}\mathbf{c}_j$, $\mathbf{k}_c = \boldsymbol{\mu}_c^{(t-1)}$ and $\mathbf{v}_j = \frac{n_c^{(t-1)}/D}{n_c^{(t)}|\Sigma|^{\beta/2}}\mathbf{c}_j$. Here $\sigma(\cdot) = \frac{1}{1+\exp(-x)}$ is the sigmoid activation. In matrix form, this yields,

$$\mathbf{Z}^{(t)} = \mathbf{V} \cdot \sigma\left(\beta\mathbf{Q}^\top\mathbf{K} - b\mathbf{1}_D\mathbf{1}_D^\top\right). \tag{18}$$

where $\mathbf{Q} = \Sigma^{-1}[\mathbf{c}_1,\ldots,\mathbf{c}_D] = \mathbf{W}_Q\mathbf{X}$, $\mathbf{K} = [\boldsymbol{\mu}_1^{(t-1)},\ldots,\boldsymbol{\mu}_D^{(t-1)}] = \mathbf{W}_K\mathbf{X}$, $\mathbf{V} = \frac{n_c^{(t-1)}/D}{n_c^{(t)}|\Sigma|^{\beta/2}}[\mathbf{c}_1,\ldots,\mathbf{c}_D] = \mathbf{W}_V\mathbf{X}$, $\mathbf{Z} = [\boldsymbol{\mu}_1^{(t)},\ldots,\boldsymbol{\mu}_D^{(t)}]$, and $b$ is a learnable bias. Here, $\mathbf{W}_Q, \mathbf{W}_K, \mathbf{W}_V$ are learnable parameters. This establishes the functional form of our proposed IB-Adapter.

**Remarks.** The core difference lies in the latent competition: the categorical structure forces channels to compete for assignments, enforcing $\sum_c q(c|j) = 1$, whereas the independent Bernoulli structure allows each channel-cluster pair to be evaluated independently. For VLA projectors, IB-Adapter is inherently more robust: uncorrelated noise channels exhibit low covariance with all semantic clusters, resulting in gate values near zero ($\sigma \approx 0$), thus effectively filtering nuisances without suppressing legitimate semantic signals.

# B. Implementation Details

## B.1. VLM Pre-training (Alignment Stage)

Since StableVLA introduces the Fused IB-Adapter projector, a hybrid architecture combining a standard MLP with our covariance-based IB-Adapter module, the projector weights differ from standard open-source checkpoints. Therefore, prior to robotic fine-tuning, we perform a Vision-Language Alignment stage to align the visual tokens produced by Fused IB-Adapter with the LLM's embedding space.

We strictly adhere to the Prismatic VLMs (Karamcheti et al., 2024) protocol, utilizing the LLaVA-LVIS4V-LRV dataset to ensure general-purpose visual reasoning capabilities. Table 4 (Top) details the Pre-training configurations.

## B.2. Robotic Fine-tuning & Baselines

Following alignment, we fine-tune the model for robotic manipulation. As described in Sec. 3.3, Fused IB-Adapter employs a dual-pathway mechanism controlled by a fusion coefficient $\lambda$ and optimized using Stochastic Pathway Dropout ($p_{\text{drop}}$).

To ensure optimal performance, we tailored hyperparameters for different benchmark suites. Comprehensive hyperparameters for both Pre-training and Fine-tuning across all benchmarks are summarized in Table 4.

**Baseline Configurations.** To ensure a fair and reproducible comparison, we align all baselines with our evaluation protocol:

- **OpenVLA (Kim et al., 2024):** We utilize the official 7B pre-trained checkpoints with standard inference settings.

- **OpenVLA-OFT (Kim et al., 2025):** We employ the officially released checkpoints.

- **VLA-Adapter (Wang et al., 2025):** For LIBERO, we use official checkpoints. For CALVIN, we re-trained the model using the official codebase under identical configurations. To ensure a strong baseline, we evaluated checkpoints spanning the convergence trajectory and reported the peak performance.[2]

- **OpenPi ($\pi_{0.5}$) (Intelligence et al., 2025):** We utilize the officially released model weights and follow the standard evaluation protocol provided by the authors.

*Table 4.* Detailed Hyperparameters for StableVLA across Pre-training and Fine-tuning stages.

| Stage I: Vision-Language Pre-training | |
| --- | --- |
| Base Components | LLM: Qwen2.5-0.5B    Vision: DINO-SigLIP (224px) |
| Optimization | Global Batch: 64    LR: 2e-5    Precision: BF16 |
| Fused IB-Adapter Params | Fusion Coeff. ($\lambda$): 0.3    Pathway Dropout ($p_{\text{drop}}$): 0.0 |

| Stage II: Robotic Fine-tuning | | | | | |
| --- | --- | --- | --- | --- | --- |
| | **LIBERO Benchmark** | | | | **CALVIN** |
| **Hyperparameter** | **Spatial** | **Goal** | **Long** | **Object** | **Benchmark** |
| Global Batch Size | 64 | 128 | 128 | 64 | 64 |
| Learning Rate | 2e-4 | 2e-4 | 2e-4 | 2e-4 | 2e-4 |
| LoRA Rank | 64 | 64 | 64 | 64 | 64 |
| *Fused IB-Adapter Specific Parameters (Ours)* | | | | | |
| Fusion Coeff. ($\lambda$) | 0.3 | 0.3 | 0.3 | 0.3 | 0.3 |
| Pathway Dropout ($p_{\text{drop}}$) | 0.3 | 0.4 | 0.0 | 0.3 | 0.3 |

---

[2]Specifically, we evaluated checkpoints at different training stages, yielding scores of 3.601, 4.14, 3.628, 3.92, and 4.097. We report the best result (4.14) to represent the baseline's upper bound capability.

# C. Detailed Experimental Results

In this appendix, we provide comprehensive quantitative results for all methods evaluated in our experiments.

## C.1. Full Comparison on LIBERO

Table 5 presents the complete comparison of all methods on the LIBERO benchmark, grouped by their training paradigm. Methods are categorized into three groups: (1) *OpenX Pretrain*, which includes models pretrained on the Open X-Embodiment dataset; (2) *OpenX + Web Co-train*, which additionally incorporates web data during pretraining; and (3) *VLM Direct FT*, which directly fine-tunes from vision-language models without robot-specific pretraining.

*Table 5.* Full comparison on LIBERO benchmark. Methods are grouped by training paradigm. We report success rate (%). **Bold**: best, underline: second best.

| Training | Method | Spatial | | | | Object | | | | Goal | | | | Long | | | |
|---|---|---|---|---|---|---|---|---|---|---|---|---|---|---|---|---|---|
| | | C | S3 | S4 | S5 | C | S3 | S4 | S5 | C | S3 | S4 | S5 | C | S3 | S4 | S5 |
| OpenX Pretrain | OpenVLA | 80.0 | 40.9 | 24.6 | 14.7 | 69.6 | 18.2 | 10.4 | 2.7 | 74.0 | 38.7 | 27.0 | 16.3 | 55.5 | 20.5 | 12.4 | 7.0 |
| | OpenVLA-OFT | 92.6 | 89.3 | 84.0 | 72.1 | 98.4 | 82.5 | 69.2 | 52.8 | 96.8 | **94.5** | 84.6 | 70.3 | **94.4** | **77.6** | 61.9 | 40.3 |
| OpenX + Web Co-train | OpenPI0.5 | **98.4** | 88.3 | 79.0 | 62.4 | **99.4** | **97.1** | **88.4** | **76.4** | 97.2 | 87.2 | 82.5 | 64.2 | 92.0 | 76.1 | **65.6** | **47.7** |
| VLM Direct FT | VLA-Adapter-Pro | 96.0 | 93.7 | 83.3 | 58.5 | 96.8 | 71.0 | 44.1 | 29.3 | 97.4 | 79.5 | 64.7 | 47.3 | 94.4 | 63.5 | 41.0 | 26.2 |
| | StableVLA | 96.2 | **94.3** | **92.1** | **82.0** | 98.8 | 92.4 | 83.6 | 70.2 | **98.0** | 93.4 | **85.0** | **71.9** | 93.6 | 76.3 | 62.4 | 45.3 |

## C.2. Per-Method Detailed Results

The following tables provide detailed per-corruption-type results for each method. Each table reports performance across all 19 corruption types (18 for tasks without Glass Blur) at severity levels 3, 4, and 5, along with clean (uncorrupted) performance.

**StableVLA (Ours).** Table 6 shows the detailed results for our method on both LIBERO and CALVIN benchmarks.

**VLA-Adapter-Pro.** Table 7 presents results for VLA-Adapter-Pro, the strongest baseline that shares our VLM direct fine-tuning paradigm.

**OpenVLA.** Table 8 shows results for the base OpenVLA model.

**OpenVLA-OFT.** Table 9 presents results for OpenVLA with orthogonal fine-tuning.

**OpenPI.** Table 10 shows results for OpenPI, which leverages internet-scale co-training.

## C.3. Ablation Study Details

The following tables provide detailed results for our ablation study on adapter architecture design.

**IB-Adapter.** Table 11 shows results for IB-Adapter, which uses only the image-bridge component without feature fusion.

**Fused IB-Adapter-softmax.** Table 12 presents results for a variant using softmax normalization instead of sigmoid in the fusion mechanism.

*Table 6.* Detailed results for StableVLA on LIBERO and CALVIN benchmarks. We report success rate (%) for LIBERO and average completed tasks for CALVIN.

| | LIBERO | | | | | | | | | | | | CALVIN | | |
| | Spatial | | | Object | | | Goal | | | Long | | | - | | |
| Corruption | S3 | S4 | S5 | S3 | S4 | S5 | S3 | S4 | S5 | S3 | S4 | S5 | S3 | S4 | S5 |
|---|---|---|---|---|---|---|---|---|---|---|---|---|---|---|---|
| Clean | 96.2 | - | - | 98.8 | - | - | 98.0 | - | - | 93.6 | - | - | 4.17 | - | - |
| Gaussian Noise | 96.0 | 92.8 | 74.0 | 95.4 | 85.0 | 64.8 | 95.2 | 85.2 | 53.2 | 82.8 | 53.6 | 25.0 | 2.02 | 1.26 | 0.60 |
| Shot Noise | 95.4 | 88.0 | 71.0 | 93.2 | 80.6 | 64.0 | 96.2 | 81.4 | 61.8 | 84.8 | 63.0 | 23.4 | 1.85 | 0.82 | 0.46 |
| Impulse Noise | 94.6 | 93.2 | 80.8 | 96.0 | 83.2 | 62.8 | 96.2 | 81.8 | 57.2 | 81.2 | 55.2 | 30.2 | 2.60 | 1.32 | 0.72 |
| Speckle Noise | 94.6 | 96.6 | 90.4 | 95.6 | 93.2 | 87.2 | 97.8 | 96.0 | 86.4 | 87.2 | 84.0 | 71.8 | 2.16 | 1.49 | 0.90 |
| Gaussian Blur | 91.6 | 83.0 | 49.4 | 84.4 | 45.4 | 2.4 | 91.8 | 74.0 | 42.6 | 55.4 | 31.2 | 2.8 | 2.18 | 0.94 | 0.34 |
| Glass Blur | 91.2 | 83.6 | 55.8 | - | - | - | - | - | - | - | - | - | - | - | - |
| Defocus Blur | 89.4 | 81.6 | 65.0 | 72.8 | 46.8 | 24.8 | 85.8 | 63.0 | 47.2 | 49.6 | 27.2 | 10.6 | 1.56 | 0.64 | 0.34 |
| Motion Blur | 89.6 | 90.4 | 82.6 | 98.2 | 72.0 | 27.4 | 93.4 | 59.8 | 37.6 | 66.0 | 21.6 | 1.0 | 1.23 | 0.46 | 0.24 |
| Zoom Blur | 95.8 | 93.6 | 86.2 | 60.8 | 49.8 | 46.8 | 91.0 | 82.8 | 69.0 | 36.8 | 16.8 | 11.2 | 3.48 | 2.86 | 2.10 |
| Fog | 95.0 | 94.0 | 94.4 | 99.8 | 99.6 | 94.8 | 94.4 | 95.2 | 81.4 | 78.4 | 75.0 | 57.0 | 2.84 | 1.97 | 0.88 |
| Frost | 96.0 | 93.8 | 89.8 | 84.0 | 79.4 | 67.4 | 92.0 | 88.6 | 81.0 | 72.6 | 69.2 | 54.8 | 3.11 | 2.99 | 2.64 |
| Snow | 95.4 | 94.8 | 94.2 | 97.4 | 92.2 | 95.4 | 92.0 | 85.4 | 95.4 | 61.0 | 39.2 | 33.0 | 3.45 | 2.50 | 1.62 |
| Spatter | 96.6 | 95.0 | 95.2 | 98.2 | 98.6 | 93.4 | 93.8 | 94.4 | 91.2 | 90.8 | 94.4 | 87.2 | 4.13 | 4.04 | 3.74 |
| Contrast | 94.2 | 95.2 | 68.0 | 99.0 | 98.4 | 78.6 | 97.6 | 95.6 | 56.8 | 93.6 | 90.2 | 52.6 | 3.74 | 2.30 | 0.59 |
| Brightness | 96.4 | 96.2 | 97.0 | 98.4 | 99.0 | 98.4 | 98.0 | 99.0 | 98.0 | 92.4 | 90.2 | 90.6 | 4.04 | 3.48 | 2.71 |
| Saturate | 96.6 | 96.4 | 98.0 | 98.8 | 98.2 | 97.2 | 96.6 | 98.2 | 97.8 | 93.4 | 92.0 | 89.8 | 4.20 | 4.19 | 4.17 |
| JPEG Comp. | 96.2 | 96.6 | 94.4 | 98.6 | 98.8 | 97.4 | 96.8 | 97.4 | 96.8 | 89.4 | 88.8 | 73.8 | 2.79 | 2.42 | 1.50 |
| Pixelate | 94.6 | 95.2 | 94.4 | 96.4 | 96.4 | 95.6 | 97.6 | 96.8 | 95.6 | 92.8 | 93.0 | 84.6 | 2.60 | 2.86 | 2.69 |
| Elastic Trans. | 91.6 | 90.4 | 78.2 | 95.6 | 89.0 | 65.8 | 74.4 | 55.2 | 45.2 | 65.2 | 38.8 | 16.2 | 1.94 | 1.49 | 0.99 |

*Table 7.* Detailed results for VLA-Adapter-Pro on LIBERO and CALVIN benchmarks. We report success rate (%) for LIBERO and average completed tasks for CALVIN.

| | LIBERO | | | | | | | | | | | | CALVIN | | |
| | Spatial | | | Object | | | Goal | | | Long | | | - | | |
| Corruption | S3 | S4 | S5 | S3 | S4 | S5 | S3 | S4 | S5 | S3 | S4 | S5 | S3 | S4 | S5 |
|---|---|---|---|---|---|---|---|---|---|---|---|---|---|---|---|
| Clean | 96.0 | - | - | 96.8 | - | - | 97.4 | - | - | 94.4 | - | - | 4.14 | - | - |
| Gaussian Noise | 98.2 | 93.8 | 32.4 | 97.2 | 53.8 | 0.0 | 79.4 | 53.4 | 26.0 | 69.4 | 37.0 | 1.0 | 2.35 | 1.03 | 0.29 |
| Shot Noise | 97.8 | 84.6 | 12.8 | 93.8 | 23.4 | 0.0 | 74.6 | 49.2 | 36.6 | 76.4 | 22.8 | 4.0 | 2.06 | 0.80 | 0.21 |
| Impulse Noise | 99.0 | 93.2 | 38.0 | 98.8 | 67.8 | 0.0 | 77.4 | 56.8 | 26.4 | 72.0 | 38.6 | 2.6 | 2.92 | 1.14 | 0.37 |
| Speckle Noise | 97.8 | 97.4 | 85.0 | 97.2 | 85.4 | 40.0 | 75.2 | 67.8 | 54.6 | 83.2 | 71.2 | 28.2 | 2.59 | 1.76 | 0.90 |
| Gaussian Blur | 89.2 | 56.2 | 3.8 | 28.0 | 0.0 | 0.0 | 87.6 | 72.6 | 26.8 | 37.0 | 3.2 | 0.0 | 0.76 | 0.12 | 0.00 |
| Glass Blur | 72.6 | 41.4 | 8.0 | - | - | - | - | - | - | - | - | - | - | - | - |
| Defocus Blur | 77.8 | 35.6 | 6.6 | 5.0 | 0.0 | 0.0 | 81.6 | 60.0 | 36.4 | 19.6 | 1.4 | 0.0 | 0.41 | 0.06 | 0.00 |
| Motion Blur | 88.0 | 59.2 | 29.0 | 11.6 | 0.0 | 0.0 | 89.2 | 47.2 | 21.4 | 36.2 | 8.0 | 0.0 | 0.69 | 0.08 | 0.01 |
| Zoom Blur | 87.8 | 78.0 | 71.8 | 19.2 | 6.0 | 0.0 | 73.2 | 59.0 | 41.0 | 9.0 | 1.2 | 2.0 | 2.95 | 2.43 | 1.76 |
| Fog | 98.6 | 98.0 | 93.6 | 63.4 | 11.8 | 0.2 | 77.4 | 71.0 | 46.2 | 52.4 | 37.0 | 12.6 | 2.77 | 2.38 | 1.34 |
| Frost | 93.8 | 90.2 | 81.4 | 32.2 | 22.8 | 8.2 | 59.8 | 55.0 | 43.2 | 49.4 | 36.4 | 19.0 | 3.61 | 3.46 | 3.37 |
| Snow | 97.2 | 91.6 | 95.6 | 63.0 | 7.6 | 59.2 | 58.2 | 40.6 | 57.0 | 41.8 | 6.0 | 13.6 | 3.64 | 3.03 | 2.73 |
| Spatter | 98.4 | 96.6 | 88.0 | 97.2 | 53.4 | 6.0 | 89.2 | 77.6 | 61.0 | 82.4 | 68.2 | 36.2 | 4.05 | 3.93 | 3.23 |
| Contrast | 99.4 | 97.8 | 25.8 | 88.6 | 0.0 | 0.0 | 91.4 | 79.0 | 26.0 | 68.6 | 4.4 | 0.0 | 2.54 | 0.78 | 0.01 |
| Brightness | 98.2 | 99.4 | 97.8 | 98.8 | 98.4 | 98.6 | 89.8 | 89.6 | 88.2 | 91.2 | 91.4 | 86.0 | 4.16 | 4.12 | 3.74 |
| Saturate | 99.2 | 99.2 | 99.2 | 98.4 | 98.4 | 94.0 | 98.0 | 92.8 | 88.0 | 93.8 | 80.8 | 84.0 | 4.13 | 4.13 | 4.14 |
| JPEG Comp. | 99.0 | 98.2 | 98.6 | 98.4 | 98.8 | 98.8 | 87.4 | 84.4 | 81.6 | 90.4 | 88.8 | 81.4 | 1.80 | 1.14 | 0.53 |
| Pixelate | 98.2 | 97.2 | 94.8 | 96.2 | 91.0 | 80.0 | 97.0 | 95.0 | 88.6 | 89.6 | 85.0 | 69.4 | 2.65 | 2.01 | 1.94 |
| Elastic Trans. | 90.2 | 74.8 | 50.2 | 90.8 | 75.0 | 42.6 | 44.2 | 14.2 | 2.8 | 80.4 | 56.8 | 31.8 | 1.89 | 1.61 | 1.26 |

*Table 8.* Detailed results for OpenVLA on LIBERO benchmark. We report success rate (%).

| | Spatial | | | Object | | | Goal | | | Long | | |
| Corruption | S3 | S4 | S5 | S3 | S4 | S5 | S3 | S4 | S5 | S3 | S4 | S5 |
|---|---|---|---|---|---|---|---|---|---|---|---|---|
| Clean | 80.0 | - | - | 69.6 | - | - | 74.0 | - | - | 55.5 | - | - |
| Gaussian Noise | 50.2 | 0.6 | 0.0 | 0.6 | 0.0 | 0.0 | 31.4 | 4.8 | 0.0 | 8.4 | 0.2 | 0.0 |
| Shot Noise | 39.0 | 0.0 | 0.0 | 0.4 | 0.0 | 0.0 | 23.6 | 2.8 | 0.0 | 5.6 | 0.0 | 0.0 |
| Impulse Noise | 55.2 | 1.4 | 0.0 | 3.0 | 0.0 | 0.0 | 41.0 | 4.8 | 0.0 | 9.2 | 1.6 | 0.0 |
| Speckle Noise | 61.0 | 26.0 | 2.8 | 5.2 | 0.2 | 0.0 | 40.8 | 17.2 | 8.4 | 17.6 | 3.8 | 0.0 |
| Gaussian Blur | 0.0 | 0.0 | 0.0 | 0.0 | 0.0 | 0.0 | 2.4 | 0.2 | 0.0 | 0.0 | 0.0 | 0.0 |
| Glass Blur | 0.0 | 0.0 | 0.0 | - | - | - | - | - | - | - | - | - |
| Defocus Blur | 0.0 | 0.0 | 0.0 | 0.0 | 0.0 | 0.0 | 0.4 | 0.2 | 0.8 | 0.0 | 0.0 | 0.0 |
| Motion Blur | 0.0 | 0.0 | 0.0 | 0.0 | 0.0 | 0.0 | 0.6 | 0.0 | 0.0 | 0.0 | 0.0 | 0.0 |
| Zoom Blur | 7.6 | 3.2 | 0.2 | 0.6 | 0.0 | 0.0 | 14.0 | 3.4 | 1.6 | 3.0 | 0.2 | 0.0 |
| Fog | 74.6 | 69.4 | 38.0 | 46.4 | 24.2 | 0.4 | 76.2 | 69.0 | 40.2 | 51.0 | 46.4 | 20.2 |
| Frost | 20.2 | 16.8 | 4.4 | 0.6 | 0.2 | 0.4 | 28.4 | 22.8 | 14.0 | 15.6 | 12.0 | 4.2 |
| Snow | 39.0 | 10.4 | 22.2 | 0.4 | 0.0 | 0.0 | 28.6 | 19.8 | 15.0 | 14.6 | 2.2 | 3.2 |
| Spatter | 65.4 | 32.4 | 3.8 | 14.2 | 0.2 | 0.0 | 50.2 | 37.0 | 16.2 | 36.0 | 12.6 | 2.2 |
| Contrast | 71.2 | 52.6 | 6.8 | 57.4 | 28.0 | 0.0 | 75.0 | 69.2 | 16.8 | 50.2 | 38.8 | 10.0 |
| Brightness | 78.4 | 73.8 | 62.8 | 64.6 | 53.2 | 29.0 | 76.0 | 73.4 | 60.8 | 52.6 | 45.6 | 41.2 |
| Saturate | 80.8 | 81.4 | 74.2 | 65.8 | 47.8 | 12.6 | 75.2 | 75.2 | 69.8 | 55.6 | 48.6 | 40.2 |
| JPEG Comp. | 72.4 | 69.8 | 56.6 | 42.6 | 29.4 | 5.8 | 73.8 | 63.2 | 45.0 | 31.8 | 10.0 | 4.2 |
| Pixelate | 60.0 | 29.0 | 7.2 | 24.8 | 3.6 | 0.0 | 53.6 | 21.2 | 5.2 | 15.4 | 1.0 | 0.0 |
| Elastic Trans. | 2.6 | 0.2 | 0.0 | 1.0 | 0.2 | 0.0 | 5.4 | 1.8 | 0.4 | 3.2 | 1.0 | 0.0 |

*Table 9.* Detailed results for OpenVLA-OFT on LIBERO benchmark. We report success rate (%).

| | Spatial | | | Object | | | Goal | | | Long | | |
|---|---|---|---|---|---|---|---|---|---|---|---|---|
| Corruption | S3 | S4 | S5 | S3 | S4 | S5 | S3 | S4 | S5 | S3 | S4 | S5 |
| Clean | 92.6 | - | - | 98.4 | - | - | 96.8 | - | - | 94.4 | - | - |
| Gaussian Noise | 90.4 | 89.2 | 67.8 | 88.6 | 56.4 | 21.2 | 94.4 | 87.2 | 56.0 | 74.6 | 47.6 | 13.8 |
| Shot Noise | 90.8 | 85.4 | 70.2 | 88.0 | 85.4 | 70.2 | 97.0 | 76.0 | 57.4 | 82.4 | 45.6 | 12.0 |
| Impulse Noise | 90.2 | 85.6 | 64.8 | 90.2 | 56.0 | 25.8 | 95.6 | 84.0 | 56.4 | 80.2 | 48.0 | 18.8 |
| Speckle Noise | 92.4 | 90.0 | 84.4 | 91.4 | 83.6 | 56.8 | 96.6 | 94.2 | 80.8 | 88.4 | 81.0 | 47.0 |
| Gaussian Blur | 86.8 | 80.6 | 46.6 | 53.0 | 35.0 | 4.6 | 96.4 | 84.8 | 48.0 | 78.0 | 38.4 | 6.2 |
| Glass Blur | 84.4 | 73.6 | 42.6 | - | - | - | - | - | - | - | - | - |
| Defocus Blur | 85.6 | 69.4 | 46.8 | 47.6 | 27.6 | 11.2 | 93.0 | 76.8 | 53.4 | 57.8 | 30.2 | 11.0 |
| Motion Blur | 77.6 | 55.6 | 39.4 | 57.4 | 20.6 | 3.8 | 89.6 | 51.6 | 27.8 | 42.6 | 16.0 | 4.6 |
| Zoom Blur | 82.2 | 71.8 | 62.6 | 70.8 | 44.0 | 30.8 | 87.4 | 77.6 | 69.6 | 54.8 | 38.2 | 13.4 |
| Fog | 90.4 | 87.8 | 73.0 | 94.0 | 85.6 | 59.2 | 96.0 | 92.0 | 63.0 | 81.6 | 71.0 | 37.8 |
| Frost | 91.6 | 89.0 | 83.0 | 84.0 | 80.8 | 83.0 | 90.6 | 83.8 | 70.2 | 69.2 | 61.4 | 54.2 |
| Snow | 92.4 | 89.2 | 93.0 | 88.2 | 80.2 | 61.4 | 89.6 | 64.6 | 83.0 | 61.6 | 35.2 | 29.0 |
| Spatter | 93.2 | 90.0 | 84.0 | 96.6 | 92.6 | 86.4 | 95.4 | 90.8 | 84.6 | 88.6 | 90.8 | 71.8 |
| Contrast | 91.8 | 89.4 | 69.6 | 96.0 | 90.6 | 56.8 | 98.4 | 93.2 | 66.2 | 91.0 | 84.6 | 32.4 |
| Brightness | 93.8 | 94.2 | 92.4 | 96.0 | 94.8 | 93.0 | 99.0 | 98.2 | 97.6 | 91.2 | 91.8 | 90.0 |
| Saturate | 92.8 | 92.4 | 92.8 | 92.8 | 95.4 | 94.0 | 95.4 | 95.0 | 97.6 | 91.6 | 91.8 | 85.2 |
| JPEG Comp. | 92.2 | 92.2 | 92.6 | 97.0 | 96.8 | 91.8 | 98.6 | 96.0 | 97.2 | 92.0 | 87.4 | 83.8 |
| Pixelate | 91.6 | 90.8 | 90.4 | 86.2 | 69.6 | 64.2 | 97.0 | 97.4 | 96.6 | 91.8 | 90.0 | 84.6 |
| Elastic Trans. | 86.6 | 80.2 | 74.4 | 66.6 | 51.2 | 36.0 | 90.6 | 79.8 | 60.2 | 80.0 | 65.2 | 29.6 |

*Table 10.* Detailed results for OpenPI on LIBERO benchmark. We report success rate (%).

| | Spatial | | | Object | | | Goal | | | Long | | |
|---|---|---|---|---|---|---|---|---|---|---|---|---|
| Corruption | S3 | S4 | S5 | S3 | S4 | S5 | S3 | S4 | S5 | S3 | S4 | S5 |
| Clean | 98.4 | - | - | 99.4 | - | - | 97.2 | - | - | 92.0 | - | - |
| Gaussian Noise | 98.4 | 92.0 | 29.8 | 98.4 | 98.0 | 58.4 | 96.2 | 78.0 | 21.6 | 88.4 | 73.2 | 13.0 |
| Shot Noise | 98.0 | 75.0 | 24.2 | 98.8 | 95.8 | 53.0 | 94.0 | 79.6 | 31.8 | 89.4 | 72.6 | 29.8 |
| Impulse Noise | 98.4 | 94.8 | 36.0 | 99.4 | 98.6 | 62.0 | 95.2 | 79.4 | 28.2 | 90.0 | 73.4 | 17.6 |
| Speckle Noise | 97.6 | 98.0 | 77.8 | 98.4 | 98.6 | 96.6 | 96.6 | 92.6 | 86.6 | 87.0 | 83.6 | 79.0 |
| Gaussian Blur | 41.4 | 10.8 | 0.0 | 88.8 | 10.0 | 0.0 | 76.4 | 44.6 | 1.6 | 11.0 | 0.4 | 0.0 |
| Glass Blur | 59.8 | 39.2 | 9.0 | - | - | - | - | - | - | - | - | - |
| Defocus Blur | 43.2 | 17.6 | 6.8 | 83.0 | 32.4 | 0.8 | 69.2 | 47.2 | 11.6 | 10.6 | 0.6 | 0.0 |
| Motion Blur | 81.6 | 38.4 | 10.8 | 96.2 | 77.8 | 34.8 | 87.0 | 55.0 | 24.4 | 62.4 | 21.6 | 1.2 |
| Zoom Blur | 80.4 | 75.8 | 74.4 | 99.4 | 98.4 | 97.4 | 89.6 | 83.2 | 77.0 | 60.0 | 0.8 | 1.0 |
| Fog | 96.8 | 92.0 | 81.0 | 98.4 | 99.0 | 97.2 | 97.0 | 96.0 | 82.4 | 87.6 | 76.4 | 50.2 |
| Frost | 95.8 | 91.0 | 77.4 | 98.8 | 98.8 | 95.4 | 85.8 | 81.6 | 72.4 | 47.6 | 74.2 | 56.2 |
| Snow | 97.2 | 91.8 | 91.4 | 99.4 | 98.2 | 98.2 | 92.8 | 77.8 | 79.8 | 89.6 | 76.0 | 40.8 |
| Spatter | 99.0 | 98.0 | 96.0 | 98.6 | 99.6 | 99.4 | 96.8 | 96.8 | 93.0 | 94.0 | 93.6 | 90.0 |
| Contrast | 98.4 | 97.0 | 94.0 | 98.6 | 99.0 | 96.0 | 98.0 | 97.2 | 93.6 | 89.6 | 88.6 | 67.4 |
| Brightness | 98.2 | 98.2 | 98.0 | 98.8 | 99.2 | 99.4 | 8.2 | 97.2 | 97.0 | 92.2 | 91.8 | 93.2 |
| Saturate | 99.2 | 99.0 | 98.4 | 97.6 | 98.2 | 98.8 | 98.4 | 97.0 | 97.2 | 92.8 | 92.2 | 94.0 |
| JPEG Comp. | 96.0 | 97.2 | 96.8 | 98.4 | 99.2 | 99.0 | 96.2 | 95.0 | 95.8 | 88.6 | 88.2 | 88.2 |
| Pixelate | 99.4 | 96.8 | 88.2 | 98.0 | 90.4 | 88.0 | 97.4 | 96.4 | 80.6 | 93.6 | 83.4 | 55.8 |
| Elastic Trans. | 99.2 | 97.8 | 96.0 | 99.6 | 99.8 | 100.0 | 94.8 | 90.4 | 81.4 | 95.0 | 90.0 | 81.6 |

*Table 11.* Detailed results for IB-Adapter on LIBERO and CALVIN benchmarks. We report success rate (%) for LIBERO and average completed tasks for CALVIN.

| | LIBERO | | | | | | | | | | | | CALVIN | | |
| | Spatial | | | Object | | | Goal | | | Long | | | - | | |
| Corruption | S3 | S4 | S5 | S3 | S4 | S5 | S3 | S4 | S5 | S3 | S4 | S5 | S3 | S4 | S5 |
|---|---|---|---|---|---|---|---|---|---|---|---|---|---|---|---|
| Clean | 97.8 | - | - | 0.97 | - | - | 97.4 | - | - | 93.8 | - | - | 1.64 | - | - |
| Gaussian Noise | 95.2 | 94.2 | 79.4 | 96.6 | 80.0 | 15.0 | 85.8 | 78.0 | 57.2 | 88.0 | 54.4 | 9.8 | 1.61 | 1.58 | 1.47 |
| Shot Noise | 96.0 | 91.4 | 76.6 | 95.4 | 67.8 | 15.0 | 86.4 | 70.6 | 62.6 | 84.8 | 56.2 | 1.7 | 1.63 | 1.61 | 1.43 |
| Impulse Noise | 97.4 | 95.0 | 84.0 | 97.4 | 86.2 | 23.4 | 89.0 | 77.0 | 55.4 | 87.0 | 54.8 | 12.4 | 1.66 | 1.60 | 1.50 |
| Speckle Noise | 96.8 | 96.2 | 91.4 | 96.6 | 97.0 | 92.2 | 84.2 | 84.2 | 77.2 | 89.4 | 80.0 | 58.4 | 1.55 | 1.54 | 1.37 |
| Gaussian Blur | 88.6 | 80.8 | 32.4 | 88.0 | 36.6 | 2.0 | 93.6 | 86.0 | 68.4 | 56.0 | 41.2 | 5.2 | 1.52 | 1.44 | 1.26 |
| Glass Blur | 89.4 | 75.4 | 43.2 | - | - | - | - | - | - | - | - | - | - | - | - |
| Defocus Blur | 88.0 | 78.6 | 45.4 | 85.2 | 35.6 | 17.4 | 89.2 | 79.6 | 68.4 | 50.6 | 37.4 | 13.6 | 1.53 | 1.46 | 1.32 |
| Motion Blur | 93.0 | 87.6 | 74.6 | 91.4 | 40.6 | 2.0 | 91.2 | 75.2 | 56.0 | 71.0 | 21.0 | 1.8 | 1.44 | 1.23 | 1.16 |
| Zoom Blur | 96.0 | 90.4 | 88.0 | 50.4 | 13.2 | 12.0 | 92.6 | 83.4 | 66.6 | 44.0 | 18.5 | 10.4 | 1.54 | 1.59 | 1.48 |
| Fog | 98.4 | 98.0 | 93.4 | 96.4 | 96.4 | 56.8 | 86.6 | 86.0 | 65.4 | 87.6 | 81.4 | 52.2 | 1.11 | 1.09 | 0.92 |
| Frost | 95.2 | 93.8 | 89.4 | 94.0 | 85.8 | 69.8 | 75.0 | 70.4 | 63.0 | 68.8 | 60.8 | 43.2 | 1.70 | 1.66 | 1.61 |
| Snow | 98.8 | 97.4 | 96.6 | 97.4 | 89.4 | 98.4 | 84.4 | 69.8 | 83.4 | 69.8 | 48.0 | 50.2 | 1.55 | 1.30 | 1.00 |
| Spatter | 97.8 | 97.8 | 98.4 | 98.0 | 97.6 | 96.8 | 93.6 | 88.0 | 79.6 | 91.8 | 92.2 | 76.8 | 1.72 | 1.58 | 1.54 |
| Contrast | 95.4 | 96.0 | 82.2 | 97.6 | 84.2 | 0.2 | 97.2 | 85.0 | 56.4 | 90.6 | 77.0 | 5.2 | 1.39 | 0.61 | 0.31 |
| Brightness | 96.6 | 97.2 | 96.4 | 97.6 | 97.2 | 96.6 | 97.4 | 97.4 | 97.4 | 90.6 | 90.8 | 91.6 | 1.44 | 1.10 | 0.91 |
| Saturate | 97.4 | 95.6 | 96.6 | 98.0 | 98.0 | 98.0 | 97.4 | 97.0 | 95.4 | 92.0 | 92.4 | 90.4 | 1.69 | 1.67 | 1.66 |
| JPEG Comp. | 96.4 | 94.4 | 89.6 | 98.2 | 98.6 | 97.6 | 95.6 | 88.0 | 90.6 | 93.0 | 90.6 | 84.6 | 1.60 | 1.55 | 1.54 |
| Pixelate | 94.6 | 94.4 | 93.6 | 98.0 | 96.0 | 95.6 | 97.8 | 95.0 | 94.0 | 91.0 | 84.4 | 78.4 | 1.57 | 1.48 | 1.58 |
| Elastic Trans. | 88.4 | 79.2 | 63.6 | 95.2 | 92.6 | 69.6 | 79.0 | 66.4 | 51.4 | 57.8 | 39.2 | 16.0 | 1.77 | 1.70 | 1.71 |

*Table 12.* Detailed results for Fused IB-Adapter-softmax on LIBERO and CALVIN benchmarks. We report success rate (%) for LIBERO and average completed tasks for CALVIN.

| | LIBERO | | | | | | | | | | | | CALVIN | | |
| | Spatial | | | Object | | | Goal | | | Long | | | - | | |
| Corruption | S3 | S4 | S5 | S3 | S4 | S5 | S3 | S4 | S5 | S3 | S4 | S5 | S3 | S4 | S5 |
|---|---|---|---|---|---|---|---|---|---|---|---|---|---|---|---|
| Clean | 72.2 | - | - | 96.0 | - | - | 97.2 | - | - | 93.2 | - | - | 0.46 | - | - |
| Gaussian Noise | 71.4 | 50.6 | 9.4 | 96.6 | 93.0 | 83.0 | 90.6 | 67.8 | 41.8 | 59.2 | 28.8 | 3.4 | 0.47 | 0.46 | 0.46 |
| Shot Noise | 65.0 | 34.0 | 6.0 | 95.8 | 90.2 | 85.6 | 90.4 | 66.2 | 45.6 | 60.4 | 20.6 | 2.6 | 0.45 | 0.47 | 0.45 |
| Impulse Noise | 68.6 | 50.0 | 11.8 | 97.8 | 93.4 | 87.0 | 90.6 | 66.0 | 44.8 | 62.0 | 29.2 | 4.2 | 0.47 | 0.46 | 0.46 |
| Speckle Noise | 70.2 | 55.8 | 36.8 | 97.2 | 96.2 | 90.4 | 93.2 | 89.6 | 75.0 | 71.8 | 56.2 | 29.4 | 0.47 | 0.47 | 0.46 |
| Gaussian Blur | 38.6 | 11.6 | 0.0 | 93.0 | 82.2 | 20.2 | 93.2 | 76.8 | 46.2 | 60.2 | 35.4 | 0.4 | 0.46 | 0.45 | 0.46 |
| Defocus Blur | 30.6 | 11.2 | 0.6 | 93.0 | 76.8 | 51.8 | 90.2 | 67.8 | 53.8 | 52.8 | 32.0 | 5.2 | 0.46 | 0.44 | 0.46 |
| Motion Blur | 49.0 | 21.8 | 6.8 | 93.6 | 69.0 | 20.2 | 92.8 | 59.2 | 39.0 | 56.4 | 10.4 | 0.2 | 0.46 | 0.46 | 0.45 |
| Zoom Blur | 29.4 | 19.8 | 7.8 | 48.0 | 5.8 | 18.0 | 80.8 | 73.0 | 56.0 | 28.8 | 10.0 | 1.4 | - | - | - |
| Fog | 67.0 | 55.6 | 25.6 | 97.8 | 98.6 | 98.6 | 95.2 | 93.2 | 75.4 | 81.2 | 66.8 | 17.8 | 0.47 | 0.46 | 0.46 |
| Frost | 58.4 | 52.4 | 44.8 | 85.0 | 80.8 | 68.0 | 71.8 | 65.8 | 54.4 | 34.0 | 29.4 | 13.0 | 0.44 | 0.47 | 0.45 |
| Snow | 53.0 | 47.6 | 52.2 | 96.2 | 85.0 | 94.8 | 86.2 | 66.6 | 80.6 | 58.8 | 28.0 | 20.4 | 0.46 | 0.47 | 0.45 |
| Spatter | 71.0 | 56.2 | 46.2 | 94.4 | 97.6 | 96.0 | 94.2 | 87.8 | 79.4 | 71.4 | 80.2 | 59.8 | 0.46 | 0.46 | 0.46 |
| Contrast | 73.0 | 55.6 | 6.0 | 97.6 | 98.2 | 4.2 | 97.6 | 95.6 | 64.6 | 85.0 | 68.0 | 8.0 | 0.46 | 0.46 | 0.47 |
| Brightness | 74.0 | 73.2 | 76.4 | 97.4 | 96.6 | 94.4 | 98.2 | 96.8 | 97.6 | 87.6 | 89.0 | 89.4 | 0.44 | 0.46 | 0.45 |
| Saturate | 71.8 | 70.6 | 64.6 | 98.0 | 96.6 | 96.4 | 95.6 | 97.0 | 97.2 | 91.4 | 88.6 | 84.8 | 0.46 | 0.45 | 0.46 |
| JPEG Comp. | 74.0 | 70.4 | 71.4 | 97.8 | 98.2 | 97.4 | 97.4 | 95.8 | 96.0 | 79.6 | 80.4 | 71.0 | 0.46 | 0.46 | 0.46 |
| Pixelate | 61.8 | 53.8 | 52.2 | 96.4 | 94.2 | 93.8 | 97.6 | 96.6 | 95.8 | 86.8 | 77.2 | 69.6 | 0.48 | 0.47 | 0.45 |
| Elastic Trans. | 30.6 | 19.4 | 11.8 | 94.2 | 91.6 | 68.2 | 68.0 | 40.8 | 14.4 | 47.0 | 19.8 | 0.4 | 0.45 | 0.45 | 0.44 |

## D. Radar Chart Details

Figure 5 shows normalized robustness scores across 18 corruption types. For each corruption type, scores are averaged over clean images and severity levels 3–5, then normalized such that the best method equals 1. The corruption type indices are listed in Table 13.

*Table 13.* Corruption type indices for radar charts.

| Index | Corruption Type | Index | Corruption Type |
|-------|-----------------|-------|-----------------|
| 1 | Gaussian Noise | 10 | Frost |
| 2 | Shot Noise | 11 | Snow |
| 3 | Impulse Noise | 12 | Spatter |
| 4 | Speckle Noise | 13 | Contrast |
| 5 | Gaussian Blur | 14 | Brightness |
| 6 | Defocus Blur | 15 | Saturate |
| 7 | Motion Blur | 16 | JPEG Compression |
| 8 | Zoom Blur | 17 | Pixelate |
| 9 | Fog | 18 | Elastic Transform |

*LIBERO-Spatial includes Glass Blur as index 19.

## E. Qualitative Results of Real-world Experiments

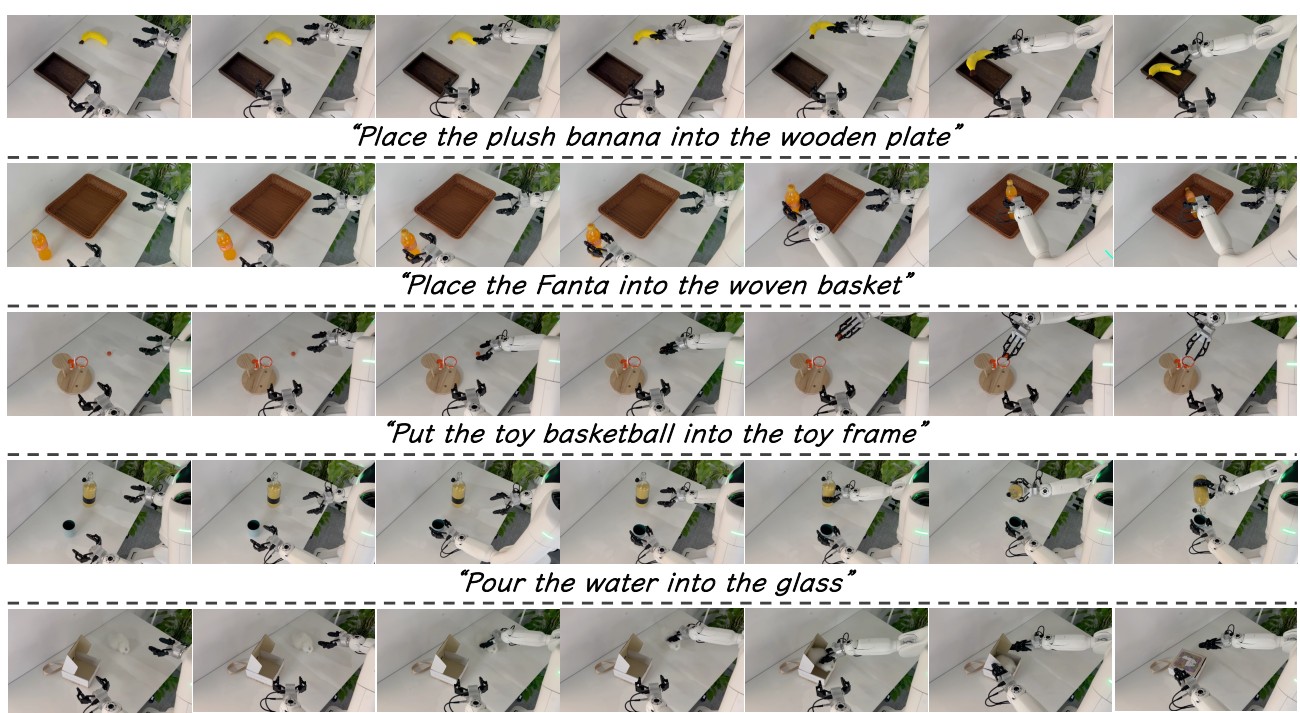

*"Place the plush banana into the wooden plate"*

*"Place the Fanta into the woven basket"*

*"Put the toy basketball into the toy frame"*

*"Pour the water into the glass"*

*"Pack the doll into the box"*

*Figure 8.* **Qualitative results of real-world experiments.** The figure displays successful execution sequences for five different manipulation tasks performed by the robot using StableVLA.

# F. Detailed Related Work and Preliminaries

## F.1. Vision-Language-Action (VLA) Models

Leveraging pre-trained Vision-Language Models (VLMs) (Liu et al., 2023b; Comanici et al., 2025; Liu et al., 2024; Bai et al., 2025; Zhu et al., 2025; Xie et al., 2024; Li et al., 2024) for robotic control has emerged as a dominant paradigm in embodied intelligence (Brohan et al., 2023; Zitkovich et al., 2023; Kim et al., 2024; Team et al., 2024; Zhang et al., 2025). However, pre-training these models from scratch relies on massive datasets, such as Open X-Embodiment (O'Neill et al., 2024) and AgiBot (contributors, 2024), requiring substantial computational resources. To mitigate this computational burden, VLA-Adapter (Wang et al., 2025) proposes a resource-efficient alternative architecture. Diverging from standard paradigms, it bypasses the expensive pre-training stage on large-scale datasets, thereby directly transferring the general perceptual capabilities of VLMs to specific robotic domains.

Despite advances in training efficiency, a critical gap remains in *architectural robustness*. In standard VLA models, the vision encoder (Zhai et al., 2023; Oquab et al., 2023) is typically frozen during end-to-end training to preserve semantic priors (Kim et al., 2024; 2025), meaning input-level noise or corruption is propagated through the visual backbone. To align visual features with the downstream policy's action space, existing models use simple MLP projectors, which ideally act as the interface to suppress disturbances before they affect the policy network. Standard MLPs, while efficient at preserving spatial details, lack intrinsic mechanisms to filter out task-irrelevant nuisances. Our work addresses this by redesigning the projector's architecture to enable noise suppression during modality alignment.

## F.2. Robustness in Vision and Robotics

Perceptual robustness is critical for the reliable deployment of robotic policies. In computer vision, this is typically evaluated using benchmarks such as ImageNet-C (Hendrycks & Dietterich, 2019), which introduces visual perturbations such as noise, blur, weather and digital corruptions. While classification tasks utilize the Mean Classification Error (mCE) as a standard metric, robustness in the VLA context is fundamentally tied to policy success rate under similar perturbations. Mainstream strategies to enhance robustness primarily rely on data-augmentation. In computer vision, techniques like (Hendrycks et al., 2021) simulate corruptions during training to improve stability. Similarly, in robotic learning, Domain Randomization (Tobin et al., 2017) is the standard approach, which randomly perturbs visual textures or physical parameters in simulation.

However, these data-centric approaches face two significant limitations. First, they incur substantial training cost, often requiring models to be trained on vast augmented datasets. Second, these methods often rely on memorizing specific noise patterns, making it difficult to generalize to unseen corruption types. Therefore, we propose **StableVLA**, which focuses on intrinsic robustness through architectural design. We demonstrate that by reconstructing the modality alignment interface with the Information Bottleneck principle, VLA models can effectively filter visual perturbance without the need for exhaustive noise-pattern simulation.

## F.3. Attention Mechanism from the Perspective of Information Bottleneck

Compared to CNNs, Vision Transformers (ViTs) exhibit superior robustness against various corruptions (Bai et al., 2021; Paul & Chen, 2022). (Zhou et al., 2022) attributes this property to the self-attention mechanism, which promotes *visual grouping* where tokens aggregate into semantic clusters. This phenomenon is theoretically grounded in the Information Bottleneck (IB) principle (Tishby et al., 2000), which optimizes the trade-off between input compression and relevant information preservation. Notably, (Zhou et al., 2022) proves that under Gaussian assumptions, the iterative optimization of the IB objective is mathematically equivalent to the self-attention operation.

While standard attention operates spatially, recent works explore visual grouping across channels. XCiT (Ali et al., 2021) introduced Cross-Covariance Attention to compute channel-wise interactions, significantly reducing computational complexity. FAN (Zhou et al., 2022) further establishes that this mechanism acts as subspace clustering; by applying the IB principle to the channel dimension, the model identifies coherent semantic subspaces while suppressing noisy channels. **StableVLA** extends this insight to VLA modality alignment, integrating a multi-head covariance mechanism to filter noisy channels and ensure robust semantic propagation for embodied decision-making.

