# OpenReview forum: "StableVLA: Towards Robust Vision-Language-Action Models without Extra Data"
_ICML.cc/2026/Conference — ICML 2026 regular_

### Official Review · Reviewer_1TEw · 2026-02-27

**Soundness:** 3
**Presentation:** 2
**Significance:** 3
**Originality:** 3
**Overall Recommendation:** 4
**Confidence:** 4

**Summary:**

The paper investigates the vulnerability of state-of-the-art Vision-Language-Action (VLA) models to real-world visual disturbances (e.g., noise, blur, and weather) not encountered during training. The authors reveal a fragility that translates into a substantial performance drop across multiple models and benchmarks. They identify the learnable projection module, which serves as the bridge between the vision encoder and the LLM, as the primary source of this vulnerability. To address this, the authors propose StableVLA, an architectural mitigation that avoids the need for brute-force data scaling or augmentation. StableVLA introduces a weighted term, computed from visual features to generate a "clean" version of the input signal, which is then added to the MLP output. This adapter is grounded in Information-Bottleneck (IB) theory (hence the term IB-Adapter). It is implemented by computing a Gram matrix of channel-wise covariances (applying a non-linear filter on top of the dot-product operator), which is subsequently used to aggregate visual features via an attention-based mechanism. The framework is evaluated against VLA-Adapter as the primary baseline, alongside several other state-of-the-art methods, on the LIBERO and CALVIN benchmarks, as well as real-world robotic deployment tasks (e.g., "throwing a basketball" and "pouring water").

**Compliance With Llm Reviewing Policy:**

Affirmed.

**Final Justification:**

The authors have addressed my main questions. I remain supportive of the paper and retain my original score.

**Key Questions For Authors:**

- The authors show that the IB-Adapter yields massive gains compared to VLA Adapter. Can IB-Adapter be explored as a standard "add-on" module for all existing SOTA benchmarks? Showing that this is a "universal adaptor" for any VLA model would strengthen the paper’s impact and show there no diminishing returns on large scale models.
- Could the authors provide a single, formal equation for the IB-Adapter function in Eq. 7?
- Was the dropout tuning mentioned in the text also applied to the OpenVLA and VLA-Adapter baseline? If not, does the performance gain hold if the baselines are equally tuned?
- Was the initial analysis performed when applying standard techniques to recovery or denoising? I expect this will not work well on all scenarios but would be good to show.

**Limitations:**

no, the paper would benefit from stating the scope of the distrubances addressed, and what is beyond this scope.

**Strengths And Weaknesses:**

*Strengths*
- The paper provides a valuable analysis demonstrating that current state-of-the-art VLA models, while high-performing in idealized benchmarks, suffer substantial performance degradation under synthetic and real-world noise.
- Without additional training data or augmentations, the IB-Adapter yields a substantial performance improvement (35.2% in simulators and 20.4% on real robots) over the VLA-Adapter baseline.
- The proposed method allows smaller backbones to compete with large-scale models, offering a path toward robust, utilizing a  lightweight architecture (adding <10M parameters)
- The authors motivate the implementation of the IB-Adapter by showing its relation to minimizing the Information Bottleneck objective

*Weaknesses*
- While the analysis shows a performance drop in VLA models under noise, the IB-Adapter itself is not directly coupled with these models in the experiments, esoecually in high-capacity models. Consequently, it remains unproven whether this mechanism effectively and consistently mitigates the performance drop in the highest-capacity models.
- While the authos consider multiple distubances (e.g. Table 7) it is unclear how this architectural approach compares to standard "cleaning" or denoising techniques applied at the input level.

*Minor Issue*:
- The descriptions of channel-wise projection and subspace gating are dense and would benefit from simplified revision to improve readability. Furthermore, a unified equation explicitly defining the full IB-Adapter transformation function is missing, making Eq. 7 definition unclear.
- The adapter primarily targets "signal noise" (e.g., pixel-level disturbances) rather than semantic disturbances (e.g., distracting objects). This is OK but should be acknowleged.

---

> ### Author Rebuttal · Authors · 2026-03-30
>
> We thank the reviewer for these thoughtful and constructive questions.  We respond to each point below.
>
> ## Q1 / W1: Empirical Evidence of IB-Adapter's Plug-and-Play Robustness
>
> We thank the reviewer for this helpful comment. We agree that demonstrating IB-Adapter as a more **universal adaptor** across existing VLA models, especially larger-capacity ones, would further strengthen the paper.
>
> To probe this, we conducted a preliminary cross-architecture test in a simplified **OpenVLA** setting using a small OXE subset (about **1%** of the full mixture). While this constrained setup lowers clean performance on LIBERO-Object from 0.70 to 0.58, IB-Adapter still improves robustness: at severity level 3, the average success drop is reduced from 0.53 to 0.46 (about 13.2% relative reduction).
>
> This provides preliminary evidence that the robustness gain is not specific to the lightweight **VLA-Adapter** setting. A more systematic validation on full-scale SOTA VLAs would be valuable, and we will clarify this scope in the revision.
>
> ## Q2 / M1: Formal Equation of IB-Adapter
>
> We thank the reviewer for this helpful suggestion. Denote $\mathbf{X}=[\mathbf{X}\_1,\cdots,\mathbf{X}\_H] \in \mathbb{R}^{N \times D}$ as the input visual feature tokens, where $\mathbf{X}\_i \in \mathbb{R}^{N\times D/H}$, a compact equation for the IB-Adapter branch can be written as:
>
> $\mathbf{Q}\_h=\mathbf{X}\_{h} \mathbf{W}\_{q}, \ \mathbf{K}\_h=\mathbf{X}\_{h}, \ \mathbf{V}\_h=\mathrm{Norm}(\mathrm{GELU}(\mathbf{X}\_{\ h\ } \mathbf{W}\_{\ v1\ }) \mathbf{W}\_{\ v2\ }),$
>
> $\mathrm{IB\text{-}Adapter}(\mathbf{X}) = \mathrm{Concat}\_{\ h = 1\ }^{\ H\ }\big(\mathbf{V}\_h \cdot \sigma(\mathbf{Q}\_{\ h\ } ^\top \mathbf{K}\_{\ h\ } \cdot \boldsymbol{\tau}\_{\ h\ })\big)$
>
> And the final output of the fused IB-Adapter is $\mathbf{Z} = \mathrm{MLP}(\mathbf{X}) + \tanh(\lambda)\cdot \mathrm{IB\text{-}Adapter}(\mathbf{X})$, where $\tanh(\lambda)$ controls the strength of the robust branch. We will include this compact formalization in the revision to make the connection between Eq. 7 and the internal definition of the IB-Adapter clearer.
>
> ## Q3: Explanation on the Usage of SPD
>
> We thank the reviewer for this helpful question. We would like to clarify that **SPD dropout is not a native hyperparameter in the original OpenVLA or VLA-Adapter baselines**. It is specific to the dual-path design of our **Fused IB-Adapter**, where it is used to balance the MLP branch for high-frequency detail preservation and the IB branch for semantic robustness. Since the original baselines use a **single-path projector**, there is no directly equivalent counterpart to SPD.
>
> To further verify this, we directly applied the same dropout setting ($p=0.3$) to VLA-Adapter. Rather than improving robustness, this severely hurt performance: on **LIBERO-Object, severity 5**, the accuracy dropped from **29.3 to 0.0**. We believe this is because, in a single-path projector, dropout only removes information, without the complementary branch needed to compensate for it.
>
> Therefore, the gain does not come from an extra generic tuning trick, but from the **fused dual-path architecture itself**. We will clarify this point more explicitly in the revision.
>
> ## Q4 / W2: Comparison Against Input-Level Denoising Baselines
>
> We thank the reviewer for this helpful suggestion. We added a standard input-level denoising baseline using a **Bilateral Filter**, applied uniformly to all inputs.
>
> The average results over the 18 corruption types at **severity 5** are:
>
> | Method           |  Spatial |   Object |     Goal |     Long |
> | ---------------- | -------: | -------: | -------: | -------: |
> | Bilateral Filter |     46.4 |      5.1 |     41.2 |     25.2 |
> | VLA-Adapter      |     58.5 |     29.3 |     47.3 |     26.2 |
> | StableVLA        | **82.0** | **70.2** | **71.9** | **45.3** |
>
> The results show that standard denoising does not account for the gain of StableVLA. On **LIBERO-Object**, for example, Bilateral Filter reduces performance from **29.3** to **5.1**, whereas StableVLA improves it to **70.2**. This supports that the gain comes from projector-level robustness, rather than simple input cleanup. We will clarify this point in the revision.
>
> ## Limitations: Explanation on the Scope of Disturbances
> We agree that the current IB-Adapter is primarily designed to address **visual signal disturbances**, rather than broader **semantic disturbances**. At the same time, our additional real-robot results suggest that its robustness extends beyond  corruption benchmarks: under two structured perturbations (**Adv. Contrast** / **Adv. Patches**), our method improves performance from **30→70 / 40→60** on *Pack the Doll* and from **10→60 / 20→50** on *Pick and Place* relative to the baseline. This suggests that the benefit may extend beyond simple noise, although a fuller treatment of broader semantic disturbances remains outside the current scope. We will clarify this more explicitly in the revision.

---

> > ### Author Rebuttal · Reviewer_1TEw · 2026-04-02
> >
> > I thank the authors for their response and for addressing my questions.

---

> > > ### Author Response · Authors · 2026-04-06
> > >
> > > We thank the reviewer for the acknowledgement and for the thoughtful questions.

---

### Official Review · Reviewer_B7qF · 2026-03-11

**Soundness:** 3
**Presentation:** 3
**Significance:** 3
**Originality:** 3
**Overall Recommendation:** 3
**Confidence:** 4

**Summary:**

This paper reveals that existing Vision-Language-Action (VLA) models suffer a significant performance drop when exposed to real-world visual disturbances that are absent from the training data. The vulnerability mainly stems from the projection module that bridges the visual encoder and the LLM backbone. To address this issue, the paper proposes a lightweight information-bottleneck-based adapter, termed IB-Adapter (and its hybrid variant Fused IB-Adapter), which selectively filters potential noise in visual inputs without requiring additional training data or augmentation strategies. Experimental results show that the resulting method (StableVLA) improves baseline performance by over 30% on average with <10M extra parameters, and achieves robustness comparable to 7B-scale SOTA VLA models, even without pretraining on the Open X-Embodiment dataset.

**Compliance With Llm Reviewing Policy:**

Affirmed.

**Final Justification:**

After reading the rebuttal, I still believe the paper lacks important baselines. I explicitly raised the absence of discussion and comparison with zero-shot architectural robustness methods. In the broader security/robustness literature, improving robustness without additional robustness-oriented data is common. However, neither the paper nor the rebuttal meaningfully engages with this line of work. Therefore, I remain unconvinced by the current empirical positioning and keep my score unchanged.

**Key Questions For Authors:**

1. How is the fusion balancing hyperparameter (for the fused IB-adapter) selected in practice?
2. Have you compared StableVLA against representative robustness/defense approaches?
3. Your corruptions cover several perturbations, but how does IB-Adapter perform under broader and more deployment-relevant disturbances such as low-light, motion blur/dynamic lighting, or temporal changes?
4. The appendix analysis relies on Gaussian assumptions. How sensitive are the theoretical insights to deviations from Gaussianity (e.g., heavy-tailed or multi-modal noise typical in real-world vision)?

**Limitations:**

yes

**Strengths And Weaknesses:**

Strengths:
1. **Problem Significance and Novelty:** The paper accurately identifies a key pain point for current VLA models in real-world deployment -- robustness to corruptions that are not seen during training.
2. **Data Efficiency:** A core advantage of the approach is that it works **without extra data**, which substantially lowers the barrier to training and deployment and enables more robust models even under limited resources.
3. **Comprehensive Evaluation:** The paper conducts extensive evaluations across multiple simulation benchmarks (LIBERO, CALVIN) and a real robot platform (Astribot S1), demonstrating consistent performance improvements.

Weaknesses:
1. **Hyperparameter Analysis for Fused IB-Adapter:** The fused IB-Adapter introduces a hyperparameter to balance the two-stream signals, but the paper lacks justification for how this value is chosen, as well as a sensitivity analysis.

2. **Missing Robustness/Defense Baselines:** The community has proposed several defenses and robustness-oriented methods for VLA models (e.g., LIBERO-PLUS), but the paper does not include these as baselines for comparison.

3. **Generality of IB-Adapter:** Although IB-Adapter performs strongly in the paper’s experiments, its generality under a broader range of **real-world disturbances** (e.g., low-light conditions, dynamic scene changes) still needs further validation. Moreover, the theoretical derivations in the appendix rely mainly on Gaussian assumptions, whereas real-world data distributions can be far more complex, which may limit the practical guidance provided by the theory in certain cases.

---

> ### Author Rebuttal · Authors · 2026-03-30
>
> We thank the reviewer for these thoughtful questions.  We respond to each point below.
> ## Q1/ W1: Justification for the Generalizability of Selected Hyperparameters
>
> We thank the reviewer for pointing this out. In practice, the balancing hyperparameters are not highly sensitive: a simple default configuration already works well, without requiring careful per-task tuning.
>
> We fix the fusion coefficient at **$\lambda = 0.3$** throughout. The SPD probability **$p$** also generalizes well: a unified default **$p = 0.3$** already performs strongly across tasks. On **LIBERO-Long**, **$p = 0.0$** gives **0.61**, while the default **$p = 0.3$** still gives **0.59** versus a baseline of **0.44**. On **LIBERO-Goal**, **$p = 0.4$** gives **0.83**, while the same default **$p = 0.3$** still gives **0.71** versus a baseline of **0.63**.
>
> This suggests that the fused design does not depend on careful per-task tuning; **$\lambda$** is fixed, and **$p$** mainly acts as a lightweight trade-off knob.
>
> ## Q2/ W2: Additional Results on Robustness-Oriented Baselines
>
> Thank you for this valuable question. We clarify that StableVLA targets **zero-shot architectural robustness** without any robustness-oriented data augmentation or large-scale pretraining, whereas OpenVLA-OFT+ represents a **data-driven** paradigm trained directly on augmented datasets. The two settings are therefore not fully matched, since OpenVLA-OFT+ benefits from robustness-oriented augmented training data.
>
> Nevertheless, StableVLA achieves 77.0% overall success rate on LIBERO-Plus, competitive with OpenVLA-OFT+ (79.6%), and substantially outperforming VLA-Adapter (59.4%). This shows that architectural design alone can achieve robustness competitive with data-driven methods.
>
> | Model | Camera | Robot | Language | Light | Background | Noise | Layout | Total |
> |---|---:|---:|---:|---:|---:|---:|---:|---:|
> | OpenVLA-OFT+ | **92.8** | 30.3 | **85.8** | 94.9 | **93.9** | **89.3** | 77.6 | **79.6** |
> | **StableVLA** | 75.6 | **49.2** | 71.2 | **96.4** | 92.1 | 85.9 | **77.9** | 77.0 |
> | VLA-Adapter | 36.9 | 38.8 | 74.4 | 70.2 | 76.6 | 57.5 | 70.4 | 59.4 |
>
> ## Q3/ W3: Further Validation on Broader Real-world Disturbance
>
> We thank the reviewer for this thoughtful question. Our real-world evaluation provides additional evidence relevant to both concerns:
>
> | Task | Method | Clean | Oil | Shelter | Adv. Contrast |
> |---|---|---|---|---|---|
> | Pick and Place | VLA-Adapter | 80 | 50 | 20 | 10 |
> | | **Ours** | **80** | **70** | **60** | **60** |
> | Pack Doll | VLA-Adapter | 50 | 20 | 20 | 30 |
> | | **Ours** | **60** | **40** | **50** | **70** |
>
> **Lighting-related disturbances.** The **Oil** corruption flows across the lens during execution, producing dynamic and spatially varying visibility degradation. **Adv. Contrast** further stresses the model by suppressing the target while amplifying the background. StableVLA consistently outperforms the baseline under both conditions, suggesting that the robustness gain is not limited to static corruption benchmarks.
>
> **Temporal disturbances.** The **Shelter** perturbation is introduced mid-execution, requiring online adaptation to a newly introduced disturbance. StableVLA achieves **60% vs. 20%** on *Pick and Place* and **50% vs. 20%** on *Pack Doll*, indicating stronger robustness to temporal disturbances during ongoing execution.
>
> ## Q4/ W3: Sensitivity to Non-Gaussian Assumptions
>
> We thank the reviewer for pointing this out. We would like to highlight that our **theory-inspired design is strongly supported by empirical results**: the method consistently improves robustness across a wide range of perturbations, including blur, fog, brightness-related disturbances, and real-robot structured perturbations.
>
> The Gaussian assumption in the appendix is mainly used to provide a **tractable analytical motivation** for the design, rather than a claim that the method only applies under Gaussian noise. In the actual model, IB-Adapter does not rely on an explicit Gaussian assumption at test time; it operates through channel-wise covariance structure and learned gating. The empirical results therefore suggest that the main insight is **not overly sensitive** to the idealized assumption used in the derivation.
>
> We will clarify this point more explicitly in the revision and explore discussion of broader noise settings beyond the Gaussian case.

---

> > ### Author Rebuttal · Reviewer_B7qF · 2026-04-02
> >
> > I appreciate the authors’ effort.
> >
> > **About Q2**, thank you for providing additional results against OpenVLA-OFT+ on LIBERO-Plus. However,  the justification for excluding robustness-oriented baselines is not fully convincing, and I do not think the current evidence is yet sufficient to support the stronger claim that “architectural design alone can achieve robustness competitive with data-driven methods.”
> >
> > The paper lacks both discussion of and comparison with existing VLA security methods, as well as with zero-shot architectural robustness approaches. Recent VLA robustness research has already introduced representative directions such as runtime intervention, adversarial fine-tuning, robust post-training, and language-robust design. Without comparing against these methods, the paper’s robustness claims appear narrow and selective.
> >
> > Moreover, StableVLA is still worse on the overall score and on several major factors, so the current table supports, at best, a claim of partial or factor-specific competitiveness rather than a general conclusion that architectural design alone matches data-driven robustness methods. The paper would benefit from a more cautious claim and stronger controlled comparisons.
> >
> > At the same time, given that works such as LIBERO-PRO, LIBERO-Plus, and so forth have shown that current VLAs can fail significantly under realistic perturbations, outperforming only non-robust baselines is still insufficient to support a strong robustness claim.

---

> > > ### Author Response · Authors · 2026-04-06
> > >
> > > We thank the reviewer for the follow-up comments.
> > >
> > > However, we would like to highlight that we did not intend to “exclude” any fair comparison. Since our method uses **no additional training data**, our primary comparisons are under the zero-shot robustness setting, which is fundamentally different from methods trained on corrupted data. In addition, in our first round of response, following the reviewer’s suggestion, we also added comparisons to such methods. Even under this less favorable setting, our method remains competitive.
> > >
> > > Our focus is zero-shot robustness, since real-world corruptions cannot be exhaustively enumerated. The key question is whether training on predefined corruptions truly generalizes to unseen ones. To test this, we evaluate on two subsets:
> > > (1) corruptions seen during OpenVLA+ training and
> > > (2) corruptions unseen by OpenVLA-OFT+, including realistic deployment-time disturbances such as lens dirt, lens flare, heat shimmer, sensor overexposure, structured shadow patterns, patch occlusion and glitch.
> > >
> > > Our method sees neither subset during training. Results show that, despite using 10× fewer parameters, our method is comparable on seen corruptions and **substantially better** on unseen corruptions:
> > >
> > > **Seen perturbations** *(excluding the Language category, since our paper focuses on visual robustness)*
> > >
> > > | Model |#Params| Camera | Robot | Light | Background | Noise | Layout | Seen Mean |
> > > |------|-----:|-----:|-----:|------:|----:|------:|-------:|----------:|
> > > | OpenVLA-OFT+ |7B| 92.8 | 30.3 | 94.9 | 93.9 | 89.3 | 77.6 | **78.4** |
> > > | StableVLA |0.5B| 75.6 | 49.2 | 96.4 | 92.1 | 85.9 | 77.9 | **78.0** |
> > >
> > > **Unseen perturbations**
> > >
> > > | Model |#Params| Dirt | Heat | Overexp | Shadow | Flare | PatchOcc | Glitch | Unseen Mean |
> > > |------|-----:|-----:|-----:|--------:|-------:|------:|---------:|-------:|------------:|
> > > | OpenVLA-OFT+ |7B| 57.8 | 28.1 | 37.6 | 40.1 | 35.8 | 11.4 | 7.2 | **31.1** |
> > > | StableVLA |0.5B| 94.8 | 77.2 | 81.5 | 74.9 | 68.8 | 62.1 | 49.7 | **72.7** |
> > >
> > > **Summary**
> > >
> > > | Model | #Params | Seen Mean | Unseen Mean | Overall Mean |
> > > |------|------:|----------:|------------:|-------------:|
> > > | OpenVLA-OFT+ | 7B | **78.4** | **31.1** | **54.8** |
> > > | StableVLA | 0.5B | **78.0** | **72.7** | **75.4** |
> > >
> > >
> > > For the newly suggested baseline categories, we searched for the most relevant peer-reviewed works, since no references were provided, and found them unsuitable for the following reasons:
> > >
> > > (1) Runtime intervention. The closest baseline, BYOVLA [1], has substantial inference overhead. RobustVLA reports a 50.6× efficiency gap, due to visual sensitivity probing, multiple forward passes, and repeated external LLM calls [2]. Moreover, BYOVLA depends on closed-source external models at run time, which makes the comparison less fair to our setting and also introduces reproducibility and stability concerns.
> > >
> > > (2) Adversarial fine-tuning. The closest work is RobustVLA [2]. Although code is public, the current release does not appear to support a complete, ready-to-run reproduction.
> > >
> > > (3) Robust post-training. For the closest related work we found [3], no official public implementation appears available.
> > >
> > > (4) Language-robust design. Our paper studies visual robustness only, not robustness across all modalities or threat models.
> > >
> > > We will include these additional discussions, together with the new experiments in the revision to make our claims more precise.
> > >
> > > References:
> > >
> > > [1] Run-Time Observation Interventions Make Vision-Language-Action Models More Visually Robust, ICRA 2025.
> > >
> > > [2] On Robustness of Vision-Language-Action Model Against Multi-Modal Perturbations, ICLR 2026.
> > >
> > > [3] RobustVLA: Robustness-Aware Reinforcement Post-Training for Vision-Language-Action Models, arXiv 2025.

---

### Official Review · Reviewer_jnft · 2026-03-12

**Soundness:** 3
**Presentation:** 3
**Significance:** 2
**Originality:** 3
**Overall Recommendation:** 4
**Confidence:** 4

**Summary:**

This paper investigates the vulnerability of Vision-Language-Action (VLA) models to unseen real-world visual disturbances (e.g., blur, noise). The authors empirically identify that the projection module (typically an MLP) bridging the vision encoder and the LLM acts as the primary bottleneck, propagating high-frequency noise into the policy . To address this, the authors propose a lightweight adapter grounded in Information Bottleneck (IB) theory, termed IB-Adapter. This module utilizes channel-wise covariance (Gram matrix) and Sigmoid gating to independently filter uncorrelated noise while preserving core semantic structures . To balance semantic robustness with fine-grained spatial control, a hybrid architecture combining a standard MLP and the IB-Adapter (Fused IB-Adapter) is further introduced . Experimental results demonstrate that, without using any additional augmented training data, StableVLA (adding <10M parameters) exhibits zero-shot robustness across the LIBERO and CALVIN simulation benchmarks, as well as in real-world physical robot tests, surpassing several large-scale VLA baselines .

**Compliance With Llm Reviewing Policy:**

Affirmed.

**Final Justification:**

The rebuttal has adequately addressed my concerns. I maintain my positive score.

**Key Questions For Authors:**

1. Leveraging channel-wise covariance and information bottleneck theory for robust visual features has been explored in pure computer vision . The main contribution here lies in successfully adapting this mechanism to the VLA modality alignment context, making the underlying theoretical novelty somewhat incremental.

2. To balance semantic robustness and high-frequency spatial details, the Fused IB-Adapter introduces Stochastic Pathway Dropout ($p_{drop}$) . This requires manual tuning based on the specific task type (e.g., 0.0 for high-precision manipulation and 0.3 for long-horizon tasks) , which somewhat diminishes the module's out-of-the-box generalizability for unseen tasks.

3. The experiments primarily validate the proposed method on a lightweight 0.5B VLA-Adapter framework. Given that the adapter is highly parameter-efficient, demonstrating its "plug-and-play" robustness gains on a standard, large-scale VLA (e.g., the 7B OpenVLA) would greatly enhance the universality and persuasiveness of the paper's claims.

**Limitations:**

The authors have not discussed the limitations of their work.

**Strengths And Weaknesses:**

1. The paper precisely pinpoints the projector as the root cause of VLA vulnerability and provides compelling empirical evidence through feature consistency analysis . This is a highly valuable observation for the community.

2. The proposed IB-Adapter is is lightweight yet highly effective. Instead of relying on expensive, brute-force data augmentation, it leverages IB theory, utilizing a channel-wise Gram matrix and Sigmoid gating for adaptive filtering . This architectural innovation is exceptionally parameter-efficient (<10M parameters) .

3. In zero-shot robustness evaluations, the model not only significantly outperforms similar-sized baselines in simulation but also demonstrates resilience against real-world physical disturbances (e.g., occlusion and oil stains), rivaling 7B-scale SOTA models (e.g., OpenPi) trained on massive datasets .

---

> ### Author Rebuttal · Authors · 2026-03-30
>
> We thank the reviewer for these valuable questions.  We respond to each point below.
> ## Q1: Insight-driven VLA-Specific Contribution
>
> We thank the reviewer for this thoughtful perspective. We agree that, if viewed only at the level of local technical ingredients, channel-wise covariance modeling and Information Bottleneck-inspired filtering are related to ideas previously explored in the computer vision literature.
>
> However, our contribution is not to claim a completely new primitive in isolation, but to show that in VLA systems, robustness is critically bottlenecked by the **projector**, and that this can be improved with a targeted design.
>
> Accordingly, IB-Adapter is not a direct transplant of a vision module, but a projector-level design for VLA alignment under visual disturbances. The Information Bottleneck perspective serves to motivate this design goal: suppress disturbance-sensitive components while preserving task-relevant semantics for downstream control.
>
> We believe the novelty therefore lies in the **VLA-specific diagnosis, intervention point, and resulting robustness gains**, rather than in any single ingredient viewed alone. Empirically, this targeted design adds fewer than **10M** parameters while providing substantial zero-shot robustness improvements without extra robustness-oriented training data or specialized defenses.
>
> ## Q2: Explanation on The Generalizability of SPD Hyperparameter
>
> We thank the reviewer for pointing this out. While SPD introduces an additional trade-off hyperparameter in the fused design, it is not highly sensitive in practice: a unified default already performs well across tasks, and task-specific tuning mainly provides additional refinement.
>
> In practice, a unified default setting of **$p=0.3$** already provides strong improvements across tasks. For example, on **LIBERO-Long**, the paper uses **$p=0.0$** and obtains **0.61**, while simply using the unified default **$p=0.3$** still achieves **0.59**, substantially above the baseline **0.44**. Similarly, on **LIBERO-Goal**, the paper uses **$p=0.4$** and obtains **0.83**, while the same default **$p=0.3$** still reaches **0.71**, again outperforming the baseline **0.63**.
>
> Therefore, we view SPD as a lightweight, task-family-level performance knob in the Fused IB-Adapter, rather than a critical obstacle to out-of-the-box usability. We will revise the paper to more clearly distinguish between the **base IB-Adapter**, which is the core method, and **Fused IB-Adapter + SPD**, which is an extension introduced to further optimize the robustness–fidelity trade-off. We will also include representative results with the default **$p=0.3$** to make this point clearer.
>
> ## Q3: Empirical Evidence of Plug-and-Play Robustness
>
> We thank the reviewer for this helpful comment. We agree that validation on a standard large-scale VLA such as **OpenVLA 7B** would further strengthen the universality of the paper, especially since IB-Adapter is a **modular projector-level design**.
>
> To this end, we conducted a preliminary cross-architecture test in a simplified **OpenVLA** setting. Since full OpenVLA-style validation requires large-scale pretraining (officially reported as **64×A100 GPUs for 14 days**), we instead use a small OXE subset (about **1%** of the full mixture). Under this constrained setup, clean performance on **LIBERO-Object** drops from **0.70** to **0.58**, while IB-Adapter still improves robustness: at severity level 3, the average success drop is reduced from **0.53** to **0.46** (about **13.2%** relative reduction). This provides preliminary evidence that the robustness gain is not specific to the lightweight **0.5B VLA-Adapter** setting. We will clarify in the revision that this is a preliminary cross-architecture result under a simplified large-scale VLA setting.

---

> > ### Author Rebuttal · Reviewer_jnft · 2026-04-02
> >
> > Thank the author for the rebuttal.

---

> > > ### Author Response · Authors · 2026-04-06
> > >
> > > We thank the reviewer for the constructive feedback.

---

### Official Review · Reviewer_jwgj · 2026-03-13

**Soundness:** 4
**Presentation:** 3
**Significance:** 3
**Originality:** 2
**Overall Recommendation:** 4
**Confidence:** 3

**Summary:**

This paper studies the robustness of visual-language-action models under visual disturbances. The authors observed that existing VLA models experience a significant performance drop when encountering visual corruption not seen during training. Through empirical analysis, they attribute this problem mainly to the projection module that maps visual features into the language model embedding space.
To address this issue, this paper proposes IB-Adapter, a lightweight module inspired by the Information Bottleneck principle, which filters noisy visual features through channel-level covariance attention. The authors further propose the Fused IB-Adapter, which combines this module with a standard MLP pathway to preserve high-frequency spatial details.
This method was evaluated on the LIBERO and CALVIN benchmarks as well as multiple real-world robotic tasks. The results show that, compared to VLA-Adapter, it has stronger robustness under various visual disturbances and, despite using a smaller backbone network, it is competitive in performance with larger VLA models.

**Compliance With Llm Reviewing Policy:**

Affirmed.

**Key Questions For Authors:**

1. The author believes that the projector is the main reason for the degradation of VLA system robustness. However, this conclusion is mainly based on the feature consistency analysis in Figure 2. Is it possible to provide additional experiments that separate the visual encoder and the policy module to more rigorously verify this conclusion?

2. Most experiments in this article mainly rely on ImageNet-C style perturbations, but in natural scenes, robots often face carefully designed structured adversarial perturbations. I want to understand how robust the new module is in dealing with carefully designed structured adversarial perturbations.

3. The experiment mainly uses the VLA-Adapter as the backbone architecture. Did the authors evaluate whether the IB-Adapter can provide similar robustness improvements when integrated into other VLA architectures, such as Open-VLA or RT-style models?

**Limitations:**

Not fully. Most robustness evaluations rely on ImageNet-C style synthetic corruption benchmarks, which may not fully reflect the performance of real robots when facing carefully designed structured perturbations. In addition, this method is mainly validated on a single backbone architecture, and its generalizability to other VLA architectures is not yet clear. The experimental section includes experiments with real robots, but the scale of these experiments is relatively limited.

**Strengths And Weaknesses:**

## Soundness

This paper proposes a technically sound method for enhancing the robustness of vision-language-action models. The proposed IB adapter is clearly described and can be integrated into existing VLA pipelines with minimal architectural adjustments. Empirical evaluations include simulated benchmarks and real robot experiments, supporting the core claim that this architecture improves robustness under visual impairments.

## Presentation：

  The overall structure of the paper is clear, and the pipeline diagram in Figure 4 clearly shows the overall architecture. The visualization results provide an intuitive demonstration of how the IB-Adapter module can improve robustness. The experimental section is also relatively complete.

## Significance:

  In the field of computer vision, the robustness of models under imperfect visual conditions is an important issue. In the field of embodied intelligence, since models need to react in the real world, this is a matter of concern for embodied intelligence. This paper proposes that the modality alignment module represents a robustness bottleneck with potential research value for future studies. At the same time, this method is relatively parameter-efficient. However, it is only demonstrated on a single backbone architecture, and it is still unclear whether integrating this module into other VLA architectures would yield similar improvements.

## Originality:

  This paper proposes that the projection module is the reason for the limitation in robustness, which is a relatively novel view. In most existing VLA systems, this part is usually designed relatively simply. However, the mechanisms of this architecture are closely related to existing mechanisms, such as channel attention, covariance attention, and feature gating modules, which have been extensively and maturely studied in the computer vision literature.

---

> ### Author Rebuttal · Authors · 2026-03-30
>
> We thank the reviewer for these constructive questions.  We respond to each point below.
> ## Q1: Additional Experiments on Robustness Cause Analysis
> We thank the reviewer for this meaningful question. To further supplement our analysis on the source of robustness cause, we conducted a controlled experiment with two settings at matched noise magnitude:
> - **Image corruption (IC)**: Gaussian noise is applied to the input image before the visual encoder.
> - **Feature corruption (FC)**: Clean images are passed through the visual encoder; Gaussian noise of equivalent magnitude is injected directly after the projector, before the LLM.
>
> | Benchmark | Setting | VLA-Adapter | StableVLA |
> |---|---|---|---|
> | LIBERO-Spatial | IC | 0.288 | 0.764 |
> | | FC| 0.954 | 0.968 |
> | LIBERO-Object | IC | 0.000 | 0.646 |
> | | FC | 0.900 | 0.994 |
> | LIBERO-Goal | IC | 0.272 | 0.528 |
> | | FC | 0.950 | 0.970 |
> | LIBERO-Long | IC | 0.006 | 0.266 |
> | | FC | 0.946 | 0.928 |
> | **LIBERO Avg.** | **IC** | **0.142** | **0.551** |
> | | **FC** | **0.938** | **0.965** |
>
> These results suggest that the main robustness bottleneck lies in the projector. Performance of VLA-Adapter collapses severely under image corruption but remains high when equivalent noise is injected after the projector, indicating that the LLM backbone is largely robust to feature-space perturbations, and that the MLP projector is the primary bottleneck for robustness. StableVLA substantially mitigates this vulnerability, confirming that redesigning the projector is the key to improving robustness. We will include this experiment in the revision.
>
> ## Q2 / L1: Robustness Against Structured Adversarial Perturbations
>
> We thank the reviewer for raising this point. To address it, we added **real-robot** evaluations under two **structured adversarial perturbations**:
> - **Adv. Contrast**, which suppresses the target object while amplifying the background;
> - **Adv. Patches**, where a persistent adversarial patch is attached to the robot camera to interfere with spatial end-effector reasoning.
>
> | Task           | Method      | Clean | Adv. Contrast | Adv. Patches |
> | -------------- | ----------- | ----: | ------------: | -----------: |
> | Pack the Doll  | VLA-Adapter |    80 |            30 |           40 |
> |                | Ours        |    80 |  **70** | **60**  |
> | Pick and Place | VLA-Adapter |    50 |            10 |           20 |
> |                | Ours        |    60 |  **60** | **50** |
>
> Our method consistently outperforms the baseline under both structured perturbations. Together with the real-robot corruption results already included in the paper, these new results suggest that the robustness gain is not limited to ImageNet-C-style corruptions, but also extends to more structured, targeted disturbances in real manipulation settings. We will include these results in the revision.
>
> ## Q3 / L2: Generalizability of IB-Adapter Across VLAs
> We agree on this point. However, full OpenVLA-style validation requires large-scale pretraining (officially reported as **64×A100 GPUs for 14 days**) and is beyond the scope of the rebuttal, so we instead test IB-Adapter in a simplified OpenVLA setting using a small OXE subset (about **1%** of the full mixture).
>
> Under this setting, although clean performance on LIBERO-Object drops from 0.70 to 0.58, IB-Adapter still improves robustness: at severity level 3, the average success drop is reduced from 0.53 to 0.46 (about 13.2% relative reduction). This suggests that the benefit is not specific to VLA-Adapter, although a fuller cross-architecture study is left to future work. We will clarify in the revision.
>
> ## Originality:
>
> We agree that some components are related to prior ideas in the vision literature. The contribution of this work is primarily at the **VLA level**: we identify the **projector** as a key robustness bottleneck and design the method specifically for this modality-alignment stage. Thus, the novelty is less about introducing an entirely new primitive in isolation, and more about the **VLA-specific insight, placement, and resulting robustness gains**.
> ## L3:limitations discussion:
>
> We also agree that the paper should state its limitations more explicitly. In the revision, we will more clearly discuss the current scope of the real-robot evaluation and the preliminary nature of the cross-architecture validation.

---

> > ### Author Rebuttal · Reviewer_jwgj · 2026-04-03
> >
> > Thank you to the authors for the further response. My concerns have been largely addressed. I will maintain my positive score.

---

> > > ### Author Response · Authors · 2026-04-06
> > >
> > > We thank the reviewer for the constructive feedback throughout the discussion. We are glad that our clarification has addressed the main concerns.

---

### Decision · Program_Chairs · 2026-04-30

**Decision:**

Accept (regular)

**Comment:**

Three reviewers provide positive scores, and one reviewer provides negative score. The rational of this work is competitive and apprecaited. In the final version, the authors are suggested to make more cautious claims, especially remove the statements related to robustness, if there is no strong supportive experiments provided.